# Dormancy-to-death transition in yeast spores occurs due to gradual loss of gene-expressing ability

Théo Maire[1,2] (iD), Tim Allertz[1,2], Max A Betjes[1,2] & Hyun Youk[1,3,4,5,*] (iD)

## Abstract

**Dormancy is colloquially considered as extending lifespan by being still. Starved yeasts form dormant spores that wake-up (germinate) when nutrients reappear but cannot germinate (die) after some time. What sets their lifespans and how they age are open questions because what processes occur—and by how much—within each dormant spore remains unclear. With single-cell-level measurements, we discovered how dormant yeast spores age and die: spores have a quantifiable gene-expressing ability during dormancy that decreases over days to months until it vanishes, causing death. Specifically, each spore has a different probability of germinating that decreases because its ability to—without nutrients—express genes decreases, as revealed by a synthetic circuit that forces GFP expression during dormancy. Decreasing amounts of molecules required for gene expression—including RNA polymerases—decreases gene-expressing ability which then decreases chances of germinating. Spores gradually lose these molecules because they are produced too slowly compared with their degradations, causing gene-expressing ability to eventually vanish and, thus, death. Our work provides a systems-level view of dormancy-to-death transition.**

**Keywords** ageing; dormancy; gene expression; germination; yeast spores
**Subject Categories** Biotechnology & Synthetic Biology; Microbiology, Virology & Host Pathogen Interaction
**Mol Syst Biol. (2020) 16: e9245**

## Introduction

When starved of nutrients, microbes can become dormant by forming spores (Rittershaus *et al*, 2013; Neiman, 2011; Stragier & Losick, 1996; Errington, 2003; Lennon & Jones, 2011; Nachman *et al*, 2007; Chu *et al*, 1998; Eldar *et al*, 2009; Suel *et al*, 2006; Suel *et al*, 2007).

Dormancy is a state in which gene expression, metabolism, and all other cellular processes are thought to have nearly or completely ceased. But exactly to what extent each process of life has halted inside dormant spores remain ambiguous as attaching firm numbers to them has been challenging due to the barely detectable levels of activities inside dormant spores. Entering dormancy is a multi-step process that does not necessarily end with the spore formation. In fact, after sporulation, spores of some species are not dormant right away. For example, *Bacillus subtilis* spores—a model for prokaryotic spores—undergoes a week-long "ageing" process during which they are metabolically active, and they actively degrade or produce key transcripts to tailor their dormancy to the environmental temperature during this process (Segev *et al*, 2012). The spores of budding yeast, *Saccharomyces cerevisiae*—a model for eukaryotic spores—take between a few hours to a few days to fully enter dormancy (Brengues *et al*, 2002; Thacker *et al*, 2011). During this time, the not-yet-dormant yeast spores express various genes—such as those involved in completing the spore wall—before gradually turning off gene expressions, at the end of which the spores are considered to be dormant (Brengues *et al*, 2002; Thacker *et al*, 2011). After entering dormancy, both bacterial and yeast spores are thought to have greatly reduced metabolism and vanishingly low, if any, genome-wide expression levels (Brengues *et al*, 2002; Segev *et al*, 2012; Rittershaus *et al*, 2013). Supporting this view for yeast spores is the recent discoveries of proteins and messenger RNPs (e.g. mRNAs bound to translational machineries) becoming inactive by aggregating into macromolecular structures inside dormant yeast spores and other fungal spores (Laporte *et al*, 2008; Petrovska *et al*, 2014). The resulting solid-like, glassy cytoplasm—being packed with these aggregates—greatly hinders proteins' movements and enzymatic activities (Ablett *et al*, 1999; Cowan *et al*, 2003; Dijksterhuis *et al*, 2007; Parry *et al*, 2014; Joyner *et al*, 2016; Munder *et al*, 2016). Taken together, previous findings support the widely accepted view that dormant yeast spores have nearly ceased all their activities (Rittershaus *et al*, 2013). But technical and conceptual difficulties lie in the phrase, "nearly ceased". Technically, the nearly ceased activities inside dormant yeast spores might, for instance,

1 Kavli Institute of Nanoscience, Delft, The Netherlands
2 Department of Bionanoscience, Delft University of Technology, Delft, The Netherlands
3 CIFAR, CIFAR Azrieli Global Scholars Program, Toronto, ON, Canada
4 Program in Molecular Medicine, University of Massachusetts Medical School, Worcester, MA, USA
5 Program in Systems Biology, University of Massachusetts Medical School, Worcester, MA, USA
 *Corresponding author. Tel: +1 774 455 4492; E-mail: hyun.youk@umassmed.edu

occur too slowly so that many measurement techniques would lack the required sensitivity to detect those activities, especially in individual dormant spores rather than in lysates of many spores. On the conceptual side, given that dormant yeast spores already appear still and that, as time passes without nutrients, they eventually die—meaning that nutrients can no longer revive them—an intriguing and underexplored challenge is explaining how yeast spores age over time to transition from being dormant (i.e. alive) to dead (i.e. what sets a spore's lifespan?). Addressing this question would deepen our understanding of dormancy—which remains either poorly understood or misunderstood for many species as the emerging discoveries show (Laporte *et al*, 2008; Petrovska *et al*, 2014)—and the meaning of cellular death. We sought to address this question in the context of yeast spores.

To address our question, we first reasoned that a dormant yeast spore's lifespan is likely encoded by amounts of its stored intracellular factors (e.g. specific proteins and RNAs) which are required for restarting replicative life. Their depletions below some values may cause spores to die. For dormant spores of some species, researchers have identified several intracellular factors that affect their revival (Donnini *et al*, 1988; Herman & Rine, 1997; Dworkin & Shah, 2010; Segev *et al*, 2012; Geijer *et al*, 2012; Sturm & Dworkin, 2015; Sinai *et al*, 2015; Mutlu *et al*, 2018). For example, alanine is more likely to revive *B. subtilis* spores that have more alanine dehydrogenase stored in them than spores that stored less of it (Mutlu *et al*, 2018). But it is unclear how the known and as-yet-unknown intracellular factors that affect revivability—perhaps the numbers of stored ribosomes and RNA polymerases—collectively affect a dormant yeast or bacterial spore's lifespan, which can be months or decades (Nagtzaam & Bollen, 1994) and thus vastly longer than typical biomolecules' lifetimes. A brute-force way to address this question is identifying all intracellular factors that affect spores' revival, then measuring how the amounts of all those factors may change over time within a dormant spore, and then deducing when the spore dies, thereby determining all combinations of intracellular factors and their amounts that are necessary for sustaining dormancy. But this approach remains elusive because, for one, it requires using a current snapshot of numbers (i.e. amounts of intracellular factors now) to predict a capacity for achieving a complex future behaviour—whether the dormant spore still has an ability to wake-up if nutrients were to reappear. Addressing our question requires circumventing this difficulty.

As we will show, we found a single, readily measurable, systems-level metric—ability for a dormant yeast spore to express genes—that reveals how a dormant yeast spore eventually dies. In the process, we propose here a view of dormancy as a precisely measurable, continuous quantity that a yeast spore loses over time as it approaches death. Our study begins by giving various glucose concentrations to dormant yeast spores. From this, we discovered that not all yeast spores wake-up (germinate) to re-enter a replicative life and that those that do not germinate are primed—they undergo accelerated germinations if they encounter more glucose hours-to-days later. These phenomena, which were previously unnoticed, led us to investigate why only a fraction of genetically identical, dormant yeast spores in a population germinate despite having ample glucose. To

investigate this, we used a synthetic circuit to induce, in dormant yeast spores, expression of a gene (*GFP*) that plays no role in germination. We found that one can induce GFP expression in dormant yeast spores that are in water without any nutrients (no extracellular amino acids and glucose) and that, surprisingly, the inducible GFP level in a dormant spore tells us the probability of that spore germinating for a given concentration of glucose. Strikingly, we found that dormant yeast spores, when induced, can express genes at levels that are as high as those of vegetative yeasts. To our knowledge, we provide here one of the first evidences and quantifications of active transcription in intact, dormant yeast spores at a single-cell resolution. We found that the inducible GFP level in a dormant yeast spore is strongly and positively correlated with the amounts of all three RNA polymerases—RNAP I-III—that the spore has but is virtually uncorrelated with the amount of ribosomal RNA, which is a proxy for the amount of ribosomes in spores. Having identified these links, our paper proceeds to show how key quantities—gene-expression ability, amount of RNA polymerases, and ability to germinate—decrease over time during dormancy, up until death. This led us to quantitatively determine when dormancy ends (death occurs) for a yeast spore. Putting the results together, our paper ends by formulating a new conceptual model that explains how yeast spores gradually lose dormancy and die: key molecules that are required for gene expression, including RNA polymerases, are gradually lost during dormancy because spores cannot produce them faster than they degrade—we measured firm numbers (tens of days) for these extremely slow processes. These molecules eventually reach below some threshold amounts after which the spore has permanently lost the ability to express genes, meaning that even ample glucose cannot restore the gene-expressing ability. Since, after receiving glucose, spores must express genes in order to build a new cell that will bud off the spore (i.e. germinate), the permanent loss of gene-expressing ability at the final moments of dormancy marks the spore's death.

## Results

### Not all genetically identical spores germinate despite encountering ample glucose

We began by re-examining the conventional test for determining whether a yeast spore is dormant or dead. The test involves giving ample glucose to yeast spores and then observing whether they germinate or not (Brengues *et al*, 2002; Joseph-Strauss *et al*, 2007; Fig 1A). If the spore germinates, then it is considered to have been dormant whereas if it does not, then it is considered to have been dead. This test, however, does not reveal why a spore that does not germinate is dead in the first place and when it died. For instance, the test cannot distinguish between a spore that died *while* it was being formed versus a spore that was alive (i.e. dormant) after forming but died *during* its dormancy (and why and when the death occurred). Motivated by this deficiency, we re-examined the test by asking whether all yeast spores can indeed germinate after receiving ample glucose (Fig 1A). Starving a laboratory-standard ("wild type") homozygous diploid yeasts caused them to form genetically identical, haploid spores. Specifically,

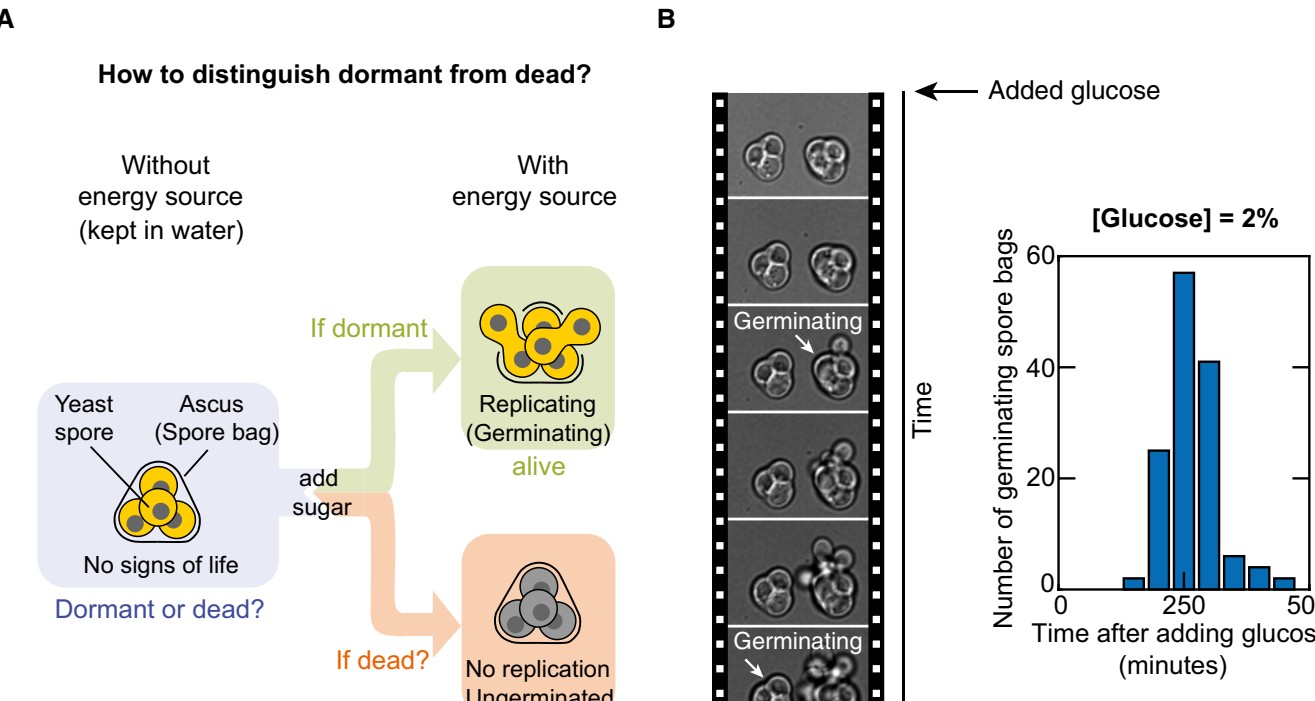

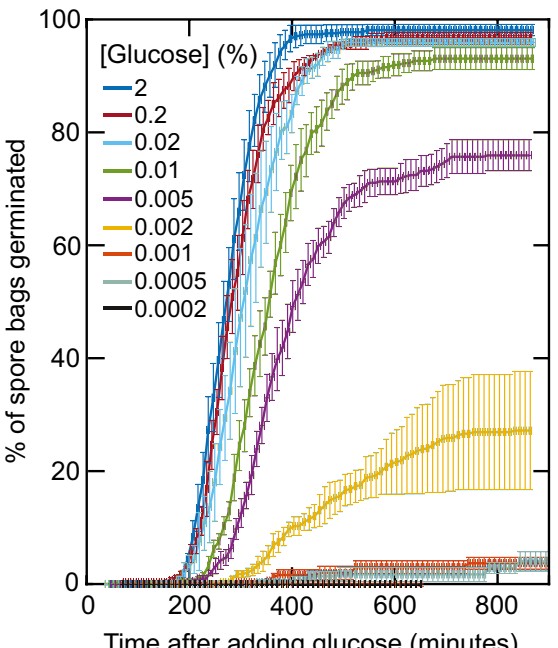

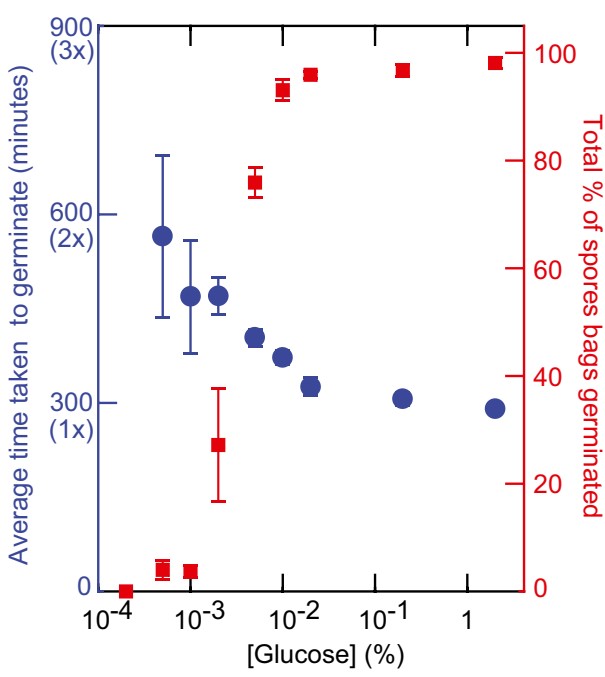

**Figure 1.**

**Figure 1. Glucose germinates only a fraction of yeast spores.**

A  Conventional test to determine whether a spore bag (i.e. ascus) containing four haploid spores, in the absence of any external nutrients (blue box), is dormant or dead. Green box: outcome if dormant. Red box: outcome if dead.

B  Left: Filmstrip of a time-lapse movie in which a 2%-glucose was added at the beginning of the movie (10 min between frames). A spore bag is counted as having germinated at the moment that at least one budding cell emerges from the spore bag (white arrows). Right: Time taken by each "wild-type" spore bag to germinate in the time-lapse movie. $n = 137$ spore bags from a representative time-lapse movie.

C  Percentage of wild-type spore bags that germinated as a function of time after adding glucose to the minimal medium. Different colours represent different glucose concentrations (from 0.0002% to 2%). $n = 3$; error bars are sem.

D  Average time taken to germinate (blue circles) and the total percentage of spore bags that germinated (red squares), both 16 h (960 min) after receiving glucose (i.e. the plateau values for each colour in Fig 1C). Both are functions of glucose concentration. $n = 3$; error bars are sem.

each diploid cell formed a single "spore bag" (i.e. ascus) that contained four genetically identical haploid spores (Fig 1A). We incubated a population of these spore bags in a "minimal medium"—this has all the essential amino acids but no glucose—and, as a modification to the conventional test, supplemented the minimal medium with a less-than-saturating concentration of glucose (less than 2%) instead of the usual, 2%-glucose. We used a wide range of glucose concentrations that spanned a 10,000-fold range, from 0.0002% to 2%. For each glucose concentration, we used a wide-field microscope to track individual spore bags and count how many of them germinated—that is, how many spore bags contained at least one spore that germinated (i.e. replicated)—as a function of time after we added glucose (Fig 1B and Appendix Fig S1). We focused on spore bags instead of individual spores that are within each spore bag because we sought to assess whether a diploid cell successfully formed at least one spore that germinates. With a saturating glucose concentration (2%), nearly every spore bag in the population germinated (Fig 1C). But with lower glucose concentrations (i.e. less than ~0.01%), a noticeable percentage of spore bags in the population—about 10% or more—did not germinate regardless of how many hours we waited after adding the glucose (Fig 1C). The percentage of spore bags that germinated changed in a sharp, step-like (sigmoidal) manner as a function of the glucose concentration (Fig 1D—red points), with the change in step located at a glucose concentration of ~0.003% (i.e. at this concentration, ~50% of spore bags germinate). In contrast, the average time taken to germinate only weakly depended on the glucose concentration, increasing by at most 2-fold despite a 10,000-fold decrease in the glucose concentration from 2% to 0.0002% (Fig 1D—blue points). This indicates that glucose weakly affects the speed of germination. Importantly, the germinations did not stop because the spores ran out of glucose for any of the glucose concentrations because when we measured how much glucose was left in the minimal medium after all the germinations stopped (i.e. ~600 min after adding glucose for all glucose concentrations), we always found a large fraction of the original glucose remaining in the medium: the glucose concentration hardly decreased even for the lowest starting glucose concentrations that we used (e.g. 0.002% and 0.001%; Appendix Fig S2). Moreover, we observed that vegetative, diploid cells of the wild-type strain—the same strain as the spores—could replicate multiple times even with the lowest glucose concentration (i.e. 0.0002%; Appendix Fig S3). Hence, even the lowest glucose concentration was ample enough for a vegetative cell to divide multiple times. These results establish that yeast spores do not necessarily germinate even when there is ample glucose.

**Spores that do not germinate after encountering ample glucose are not necessarily dead**

To explain why some spores do not germinate even with ample glucose, we considered two possibilities. One was that the spore bags that did not germinate (i.e. "un-germinated spore bags") died while trying to germinate. The other possibility was that the un-germinated spore bags were still able to germinate and thus were still alive (i.e. dormant). We distinguished these two possibilities by repeating the above experiments but now by adding glucose in two steps (Fig 2A). First, we gave the spores a relatively low concentration of glucose. We then waited, typically ~16 h (~1,000 min), after which no more germinations occurred. Then, we added more glucose to increase the total glucose concentration and then observed if this led to any more germinations. We found that some of the spore bags that did not germinate the first time were germinating after receiving more glucose (Fig 2B and Appendix Figs S4 and S5). Yet, some of the spore bags still did not germinate after receiving the second batch of glucose if the resulting, final concentration was not the saturation value, 2% (Fig 2C—top panel). For example, when the final concentration was 0.002%, nearly 60% of the spore bags in the population remained un-germinated. Intriguingly, this 60% is close to the percentage that would have remained as un-germinated if we had given the 0.002%-glucose all at once instead of in two steps (compare Fig 2C with Fig 1C). This suggests that each spore bag may be pre-programmed to germinate for certain glucose concentrations. Accordingly, nearly every spore bag eventually germinated if the second batch of glucose increased the total glucose concentration to 2% (Fig 2C—bottom panel). These results establish that spores that do not germinate after encountering ample glucose are not necessarily dead.

**Spores that do not germinate after encountering low glucose concentrations are primed so that they accelerate their germinations later**

To better understand why only some spore bags germinated for a given glucose concentration, we examined whether the un-germinated spore bags had any measurable response to the glucose that they first encountered. When we added glucose in two steps so that the final concentration was 2% (Fig 2A), we found that the spore bags took less time to germinate, in response to the second batch of glucose, than they would have if they had received the entire 2%-glucose all at once (Fig 2D). Specifically, if a spore bag was in a minimal medium without any glucose for 16 h and then encountered a 2%-glucose, it needed an average of ~200 min to germinate.

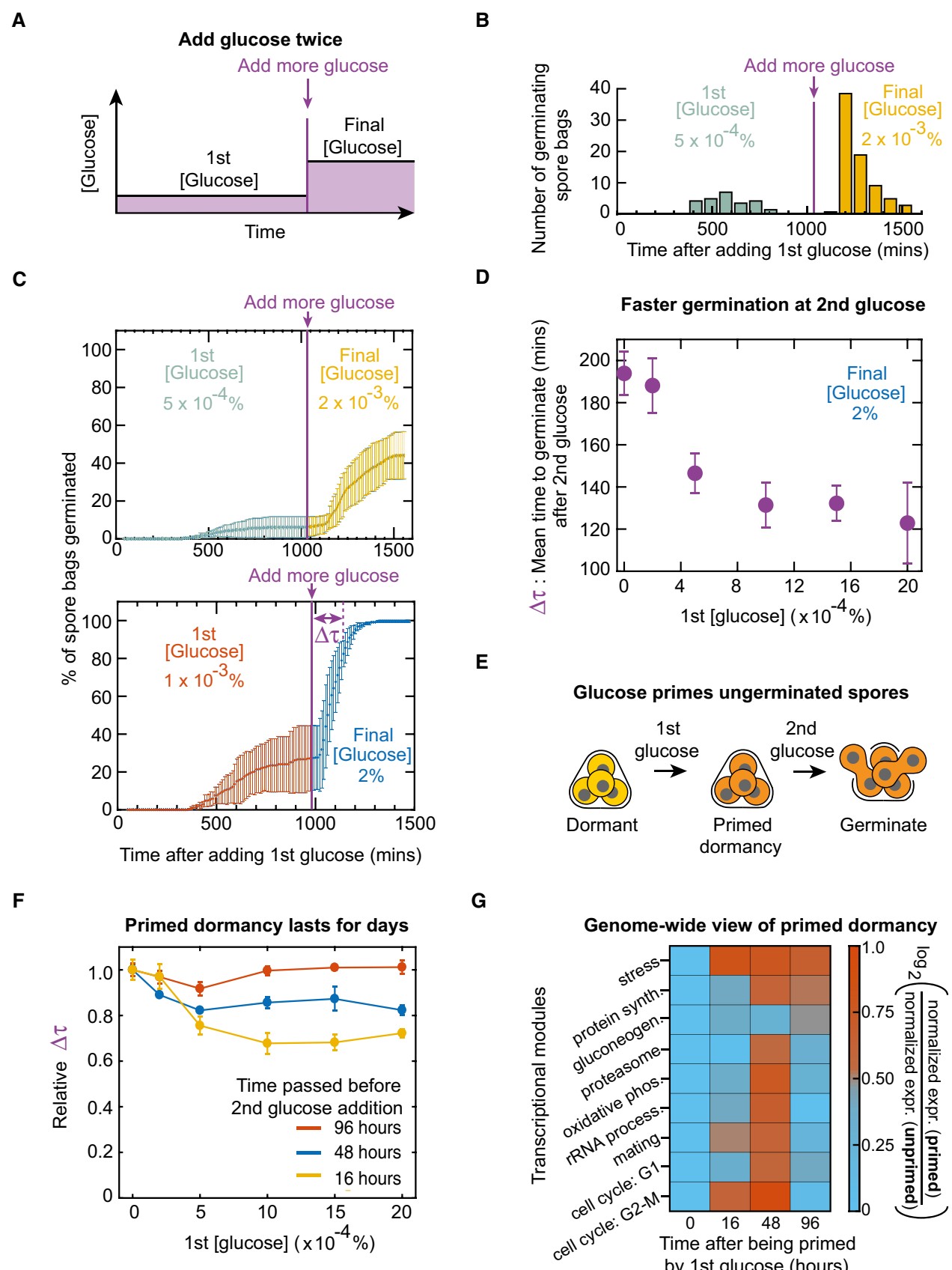

**Figure 2.**

◀

**Figure 2.   Un-germinated spores, primed by a low glucose concentration, germinate faster upon encountering more glucose a few days later.**

A  Wild-type spores are first incubated in a low glucose concentration ("1st [glucose]") before we add more glucose later to increase the glucose concentration to "final [glucose]".

B  For experiment in (A), number of spore bags germinated within each binned time. First glucose concentration is 0.0005% (from 0 to 16 h) (green bars) and the final concentration is 0.002% (from 16 to 32 h) (orange bars). $n$ = 143 spore bags (representative data).

C  For the experiment in (A), percentage of spore bags that germinated as a function of time since the first glucose addition (1st and final concentrations as indicated). More glucose added 1,000 min after the first glucose (purple vertical line). $n$ = 3; error bars are sem.

D  For the experiment in (A), average time taken for a spore bag to germinate—denoted $\Delta\tau$ in bottom panel of (C)—after stepping up the glucose concentration. We varied the first glucose concentration, but the final concentration was always 2%. $n$ = 3; error bars are sem.

E  Spore bags that do not germinate after encountering the first glucose are "primed" to germinate faster upon encountering more glucose.

F  For the experiment in (A), average time ($\Delta\tau$) taken for a spore bag to germinate after stepping up the glucose concentration (final glucose concentration is 2%), plotted as a function of the first, low glucose concentration. Different colours represent different times at which we added more glucose after the first addition: 16 h (yellow), 48 h (blue) and 96 h (red). "Relative $\Delta\tau$" is the average time $\Delta\tau$ divided by the $\Delta\tau$ of spore bags that were incubated in minimal media without any glucose (0%) for the same duration of time (after which they received a 2%-glucose). $n$ = 3; error bars are sem.

G  Heat map showing transcriptome-wide changes in un-germinated spores at 0, 16, 48 and 96 h after being primed by a 0.002%-glucose (via RNA-seq). Appendix Table S1 lists the genes for each transcriptional module (each row in the heat map). Also see Appendix Fig S8. For each transcriptional module, we first divided the expression level of each gene in that module by its expression level at 0-hours—this yields "normalized expression level" for that gene for both primed and unprimed spores (unprimed spores were in minimal media without glucose for 0, 16, 48 and 96 h). We averaged these values over all genes in a transcriptional module to obtain "normalized expression (primed)" and "normalized expression (unprimed)" for each transcriptional module. Colours represent ratio of these two values, averaged over three biological replicates ($n$ = 3).

But this time decreased by about half (i.e. to ~120 min) if a spore bag was first in a minimal medium with a low glucose concentration—ranging from 0.0002% to 0.002%—for 16 h and then received more glucose so that the final concentration was 2% (Fig 2D). Thus, encountering a very low amount of glucose "primes" some spores so that, upon encountering a saturating level of glucose at a later time, they would germinate faster—up to two times faster on average—compared with spores that did not previously encounter any glucose (Fig 2E). Furthermore, when we primed the spores with a very low glucose concentration and then waited between 16 h to 4 days before increasing the glucose concentration to 2%, the spores were still primed—they germinated faster compared with spores that were kept in minimal medium without any glucose for the same amount of time. They were primed for up to two days after we added the first glucose but no longer primed four days after we added the first glucose (Fig 2F and Appendix Fig S6). Thus, primed dormancy lasts for and decays over days.

**Transcriptome-wide view of primed dormancy**

Before turning to the question of what causes only some spore bags to germinate for a given glucose concentration, we sought to uncover gene expressions that underlie the primed dormancy. We first primed the spores by incubating them with a low glucose concentration (0.002%) for either 16 h, 1 day, 2 days or 4 days. We then used zymolyase, as is the standard (Coluccio *et al*, 2004), to isolate the un-germinated spores from the surrounding vegetative cells (Appendix Fig S7) and then analysed their transcriptomes with RNA-seq. As a control, we also analysed the transcriptome of un-primed spores, which were incubated in minimal media without glucose for the same amounts of time as the primed spores. Following an insightful previous study (Joseph-Strauss *et al*, 2007) that analysed the yeast spores' transcriptome over several hours after they received a 2%-glucose, we grouped multiple genes together into a set, called a "transcriptional module" (Ihmels *et al*, 2002; Joseph-Strauss *et al*, 2007), if those genes participate in the same process (e.g. protein synthesis; Appendix Table S1). For both the primed and un-primed spores, we averaged the expression levels of

all genes in a given module to obtain one expression value for that module. For six of nine transcriptional modules, the primed spores had higher expression values than the un-primed spores after 16 and 48 h of incubations. But both types of spores had nearly the same expression value after four days of incubation (Fig 2G—last six rows and Appendix Fig S8). This trend mirrors the trend followed by the average time taken by primed spores to germinate (i.e. accelerated germinations up to 48 h after being primed but no accelerated germinations after 4 days; Fig 2F). Two transcriptional modules showed this trend in a particularly pronounced manner. One of them is the module for mating—haploid cells mate after germinating (Fig 2G—seventh row). The other is the module for transitioning from cell cycle's G2-phase to mitosis—a crucial final step of germination (Fig 2G—last row and Appendix Fig S8). Intuitively, these results make sense since one expects that getting ready for mitosis and mating would accelerate germinations. Together, these results establish that very low glucose concentrations can trigger transcriptome-wide changes in un-germinated spores to prime them, so that they can germinate faster when they encounter more glucose hours-to-days later.

**Hypothesis on why not every yeast spore germinates with low glucose concentrations**

Although we now understand how un-germinated spores respond after receiving glucose, we have not addressed our original question: what determines, in the first place, which spore bags germinate? We hypothesized that each diploid, vegetative yeast forms a spore bag with a distinct "internal state". Many intracellular factors may define an internal state, including amounts of ATPs or ribosomes that are stored inside the spores or combinations of these and other molecules that are stored inside spores. We then hypothesized that, for each glucose concentration, only some of the spore bags have the "right" internal states that enable germination. A difficulty in testing this hypothesis is that in our experiments thus far, we gave glucose to spores *and then* observed their subsequent actions, including for discovering primed dormancy. But in such experiments, due to all measurements occurring *after* the spores receive

glucose, we cannot infer the spores' internal states that existed *before* they encountered glucose since glucose likely changes the internal states. Therefore, we sought to manipulate the internal states without giving glucose. In particular, we reasoned that depleting any internal resources (e.g. ATPs or amino acids) that are stored inside spores *before* adding glucose would either decrease—or alter in more complex ways—the percentage of spore bags that germinate for each glucose concentration.

**Synthetic circuit to induce GFP in dormant yeast spores without any nutrients**

To test our hypothesis, we built a synthetic gene-circuit in vegetative diploid yeasts so that doxycycline—a small inducer molecule that easily diffuses through the yeast's cell wall—would cause the cells to produce the Green Fluorescent Protein (GFP). In vegetative yeasts, this synthetic circuit functions in such a way that increasing the doxycycline concentration increases the GFP expression. We formed spores out of these engineered diploid cells (Fig 3A). We reasoned that if doxycycline can induce GFP expression in these spores without any nutrients (i.e. in water), then we might deplete the stored resources inside the spores and thereby alter the percentage of spore bags that germinate for a given glucose concentration. But it was unclear whether it was even possible to induce GFP expression or any arbitrary gene in dormant yeast spores without nutrients. For one, if gene induction were possible, then it is unclear why, apparently, almost all gene expressions are suppressed in dormant yeast spores (Rittershaus *et al*, 2013) since being able to induce GFP expression would mean that there must be active RNA polymerases and ribosomes, and chromosomal regions that are accessible to them. Moreover, recent studies established that starved vegetative yeasts, dormant yeast spores, and other dormant fungal spores have a solid-like, glassy cytoplasm that is packed with macroscopic aggregates of proteins and mRNPs (mRNAs bound by proteins) that inhibit gene-expression machineries and movement of molecules inside the cytoplasm (Laporte *et al*, 2008; Petrovska *et al*, 2014). By aggregating, proteins needed for metabolism and gene expression would be inactivated as well (Ablett *et al*, 1999; Cowan *et al*, 2003; Dijksterhuis *et al*, 2007; Parry *et al*, 2014; Joyner *et al*, 2016; Munder *et al*, 2016). Indeed, to our knowledge, there has not yet been a direct, single-cell-level observation of gene expression occurring in yeast spores during dormancy. A previous, bulk-level (population-level) study (Brengues *et al*, 2002) has shown, from lysates of many yeast spores, that two genes—*PGK1* (involved in gluconeogenesis) and *SPS100* (involved in forming spore walls during spore formation)—are expressed but that they turn off a few days after sporulation (i.e. while the spores enter dormancy). But it remains unclear whether their expressions completely cease or remain at low, non-zero levels during dormancy. This is because these bulk-level measurements were based on finding ribosomes bound to mRNAs after lysing populations of yeast spores, meaning that the measurements cannot clearly distinguish between zero and very low expression levels. Moreover, the ribosomes bound to mRNAs may be from the macroscopic aggregates that formed before the spores entered dormancy (Laporte *et al*, 2008; Petrovska *et al*, 2014), which may disable translation of those ribosome-bound mRNAs (Ablett *et al*, 1999; Cowan *et al*, 2003; Dijksterhuis *et al*, 2007; Parry *et al*, 2014; Joyner *et al*, 2016; Munder *et al*, 2016).

Adding to the ambiguity is the fact that a bulk-level study revealed a depletion of a minute fraction of radioactive uracil and methionine from an extracellular medium by a dense population of yeast spores (Brengues *et al*, 2002). This finding indicates that transcription (proxied by the depleted uracil) and translation (proxied by the depleted methionine) may be occurring in dormant yeast spores, though these are indirect measurements since they did not directly visualize gene-expression dynamics inside individual, intact spores. Time-lapse imaging of gene expression in individual yeast spores— as we will show with our synthetic circuit—would help in resolving these ambiguities and subtleties of gene regulations in dormant yeast spores.

Before testing the hypothesis posed in the previous section, we investigated whether GFP induction in dormant spores is even possible by incubating the engineered spores with doxycycline in either water or saline solution (PBS). Surprisingly, we discovered that doxycycline fully induced GFP expression in these spores (Fig 3B and Appendix Figs S9 and S10). Varying the doxycycline concentration tuned the spores' GFP levels over the same, wide range of values as in vegetative cells that also have the synthetic circuit (Appendix Figs S9–S11). We found that both the rate of GFP production and the final (steady state) level of GFP widely varied among spore bags of the same population (Fig 3C and Appendix Fig S12). The most striking feature was that all spore bags expressed GFP very slowly—GFP levels plateaued at steady-state values after ~20 h of doxycycline whereas the vegetative yeasts needed ~8 h of doxycycline for the GFP level to plateau (Appendix Fig S11). But a more puzzling discovery was that the spores' GFP levels stabilized at steady-state values in the first place. This is puzzling because the spores were not dividing and hence their GFP—a highly stable protein—could not be diluted away by cell divisions. In replicating (vegetative) yeasts, highly stable proteins such as GFP reach steady-state levels because their production rate matches the cell-division rate. After the spores' GFP levels reached steady-state values, we incubated the spores in PBS without doxycycline for two days during which their GFP levels remained virtually unchanged (Appendix Fig S13). This means that GFP levels reached steady-state values in dormant spores because the spores stopped producing GFP after a day, despite the saturating concentration of doxycycline (100 μg/ml) still being present in the medium after a day, with barely detectable degradations (Appendix Fig S14). Together, these results provide a direct proof, with intact spores, that one can fully activate—to the same level as in vegetative yeasts —transcription and translation of an arbitrary gene such as *GFP* in dormant yeast spores without any nutrients. These results also establish that gene expression in dormant yeast spores can exhibit starkly different dynamics when compared to vegetative yeasts, such as vastly slower timescales and gene regulations.

**Inducing GFP expression in dormant yeast spores does not alter percentages of spore bags that germinate**

Returning to our hypothesis that depleting yeast spores' internal resources through GFP induction hinders germination, we incubated spores in PBS with a saturating concentration of doxycycline (100 μg/ml) for 24 h, after which the GFP levels plateaued at steady-state values (Fig 3C and Appendix Fig S13). We then washed away the doxycycline and then incubated the spores in a minimal

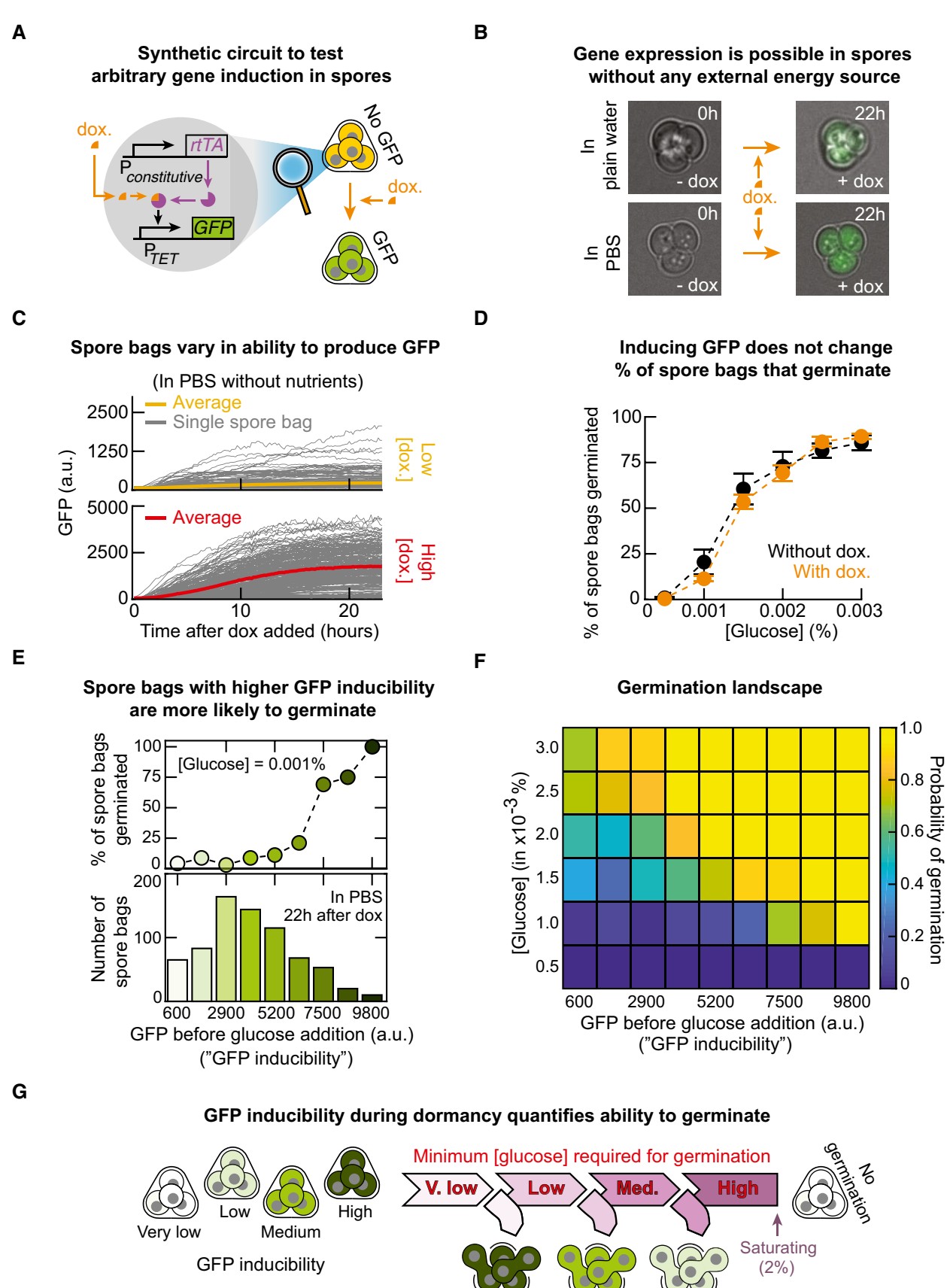

**Figure 3.**

**Figure 3.   Inducing GFP expression during dormancy with synthetic circuit shows that spores that can realize higher GFP level (higher GFP inducibility) are more likely to germinate (higher germination ability) for any glucose concentration.**

A   Synthetic gene-circuit that constitutively expresses a transcription factor, rtTA (with ADH1-promoter) and an inducible promoter (TET-promoter) controlling *GFP* expression. Increasing doxycycline increases GFP expression.

B   Engineered spore bags (shown in (A)) transcribe and translate GFP in water (without any nutrients) with 25 µg/ml of doxycycline (top row) and in a saline solution (PBS) with 50 µg/ml of doxycycline (bottom row). Snapshots of GFP expression shown 22 h after adding doxycycline.

C   GFP levels of individual spore bags (grey curves) over time (measured every 10 min with a wide-field epifluorescence microscope) after incubation in PBS with 10 µg/ml of doxycycline (top panel: $n$ = 104 spore bags) and 100 µg/ml of doxycycline (bottom panel: $n$ = 150 spore bags). (D) Engineered spore bags (shown in (A)) were first incubated for 22 h in either PBS without any doxycycline or with 100 µg/ml of doxycycline. Next, we transferred them to minimal medium with various glucose concentrations. Plot shows the total percentage of the engineered spore bags that germinated (measured 20 h after incubating with glucose) for spores pre-incubated in PBS without doxycycline (black points) and in PBS with 100 µg/ml of doxycycline (orange points). $n$ = 3; error bars are sem.

E   Top: Percentage of spore bags with the same GFP level ("GFP inducibility") from (D) that germinated after receiving a 0.001%-glucose. Percentages are averaged over all spore bags with the same binned GFP level (in the histogram below). Representative data shown ($n$ = 145 spore bags in a population).

F   Germination landscape: colours represent the probability that a spore bag with a particular steady-state GFP level ("GFP inducibility") germinates for each glucose concentration (i.e. data in top panel of (E) represents a single row of this heat map). To measure each pixel, as in the experiment described in (D), we incubated spore bags in PBS with 100 µg/ml of doxycycline for 22 h before adding glucose ([glucose] as indicated along the rows). Columns indicate GFP inducibility of a spore bag, measured at 22 h after adding the doxycycline. Each pixel is an average over three replicate populations ($n$ = 3).

G   Given a spore bag, its GFP inducibility is a read-out of the probability to germinate ("germination ability") for each glucose concentration and the minimal glucose concentration required to guarantee germination. Spore bags with lesser gene-expressing abilities without nutrients require more glucose to germinate.

medium with a fixed concentration of glucose. Then, with time-lapse microscopy, we tracked individual spore bags to determine how many spore bags germinated. For comparison, we used the same method to determine how many spore bags without GFP induction germinated—these spores were incubated for 24 h in PBS without doxycycline and then in a minimal medium with the same concentration of glucose as the GFP-induced spores (Fig 3D and Appendix Fig S15). For every glucose concentration that we examined, we found that inducing GFP expression did not appreciably alter the percentage of spore bags that germinated and that it also did not appreciably alter the average times taken by the spores to germinate (Appendix Fig S16). Thus, our hypothesis is incorrect: expressing *GFP* does not alter the spores' ability to germinate. While it may now appear that we are "back to square one"—since we still have not yet uncovered what causes only some of the spore bags to germinate for a given glucose concentration—we discovered the answer, as we explain in the next section, when we examined the GFP levels of individual spore bags rather than the average GFP level of the entire spore population as we have just done.

### GFP level that a spore can realize (GFP inducibility) encodes the probability of germinating (germination ability) for each glucose concentration

In the previous experiment, we measured the steady-state GFP levels just before the spores encountered glucose. This led to the finding that spore bags that produced more GFP—spore bags with higher "GFP inducibility"—were more likely to germinate (Fig 3E and Appendix Figs S17 and S18). For example, after encountering a 0.001%-glucose, nearly 100% of the spore bags with the highest, steady-state GFP levels (highest GFP inducibility) germinated whereas, in the same population, only ~10% of the spore bags with half the GFP inducibility germinated. In fact, for every glucose concentration, we could precisely determine the probability of germinating for a spore bag once we knew its GFP inducibility (Appendix Fig S18). We thereby established a quantitative relationship between the inducible GFP level and the ability to germinate. Importantly, this means that just because a spore bag can express any amount of GFP without nutrients does not mean that it will

germinate after receiving a certain amount of glucose. There is no binary relationship between the GFP expression and germination. The quantitative relationship between the GFP inducibility and the probability of germinating ("germination ability") for a given glucose concentration establishes that the stochastic variability (Balazsi *et al*, 2011; Padovan-Merhar & Raj, 2013) in the induced GFP expression among dormant spores is meaningful and predictive, despite GFP not having any functional role in the complex, multi-step process that is germination.

### Germination landscape visually represents germination ability as a function of glucose concentration and GFP inducibility

To visualize our results, we plotted a "germination landscape"—a heat map whose colour represents a probability that a spore bag with a given GFP inducibility germinates (i.e. germination ability) for each glucose concentration (Fig 3F and Appendix Fig S19). In the germination landscape, yellow represents a near-certain germination (i.e. germination probability of nearly 1), green represents a germination probability of ~0.5, and dark blue represents a germination probability of nearly zero. The germination landscape shows a "coastline" of nearly yellow pixels moving up toward higher rows (i.e. toward higher glucose concentrations) as one moves from right to left (i.e. as the GFP inducibility decreases), meaning that more glucose is required to guarantee a germination for a spore bag with a lesser GFP inducibility. The blue-green pixels are almost immediately below the coastline of yellow pixels, indicating that the probability of germinating, for a fixed value of GFP inducibility, is a sharp step-like function of the glucose concentration. We confirmed this by quantitatively extracting (by log-regression), from the germination landscape, the minimum glucose concentration required for a spore with a given value of GFP inducibility to have a 99%-chance of germinating (Appendix Fig S20). We call this concentration, given the sharpness of the nearly step-like probability function, the "minimum glucose concentration required for germination". We determined that as a spore bag's GFP inducibility decreases, the minimum glucose concentration required for germination increases (Fig 3G and Appendix Fig S20). Importantly, since inducing GFP expression does not alter the total percentage of spore bags that

germinate for any glucose concentration (Fig 3D), inducing GFP expression in a spore bag does not change—neither increase nor decrease—the probability of germinating. Hence, GFP inducibility merely indicates which spore bags are more likely to germinate for a given glucose concentration but does not cause or hinder germination. Hence, for the same glucose concentration, spores with higher GFP inducibility have higher germination ability than spores with lesser GFP inducibility. Moreover, spores with a higher GFP inducibility require less glucose to guarantee that they germinate (i.e. have 100% chance of germinating) than spores with lesser GFP inducibility (Fig 3G and Appendix Fig S20C).

## Transcribing within a day during dormancy is crucial for spores to remain alive

GFP has no role in germination. So, it is puzzling why GFP inducibility so precisely predicts the germination ability for each glucose concentration. To make sense of this, we reasoned that the GFP inducibility represents the dormant spore's ability to express genes in general, not just *GFP*, when they are forced to do so without any nutrients (e.g. inducers such as doxycycline force an activation of GFP expression). This is because there is nothing special about GFP. Notwithstanding the conventional view that gene expression has nearly ceased in dormant yeast spores, we then reasoned that GFP inducibility is a proxy for gene expressions that are actually occurring in dormant spores and which are required for germination. According to this reasoning, inhibiting global (genome-wide) transcription and translation during dormancy would decrease the percentages of spore bags that germinate (i.e. decrease germination ability) for each glucose concentration. To test this idea, we incubated spores in PBS for one day with one of three drugs—thiolutin, cycloheximide and antimycin A (Fig 4A). Each drug inhibits some key part of gene expression. Thiolutin globally inhibits transcription (Parker *et al*, 1991; Grigull *et al*, 2004; Guan *et al*, 2006; Pelechano & Perez-Ortin, 2008; Lauinger *et al*, 2017). Cycloheximide globally inhibits translation (Belle *et al*, 2006; Buchanan *et al*, 2016). Antimycin A inhibits oxidative phosphorylation and, in turn, synthesis of ATP—a molecule that is required for expressing *GFP* and other genes (Ocampo *et al*, 2012). After the day-long drug treatment, we washed away the drug and then added a 2%-glucose with minimal medium to check how many spores could still germinate (i.e. were still alive; Fig 4A). By convention, being a dead spore means that it cannot germinate even after receiving a 2%-glucose. This definition of death is sensible in light of the germination landscape (Fig 3F): since spores are more likely to germinate with a higher glucose concentration than with a lower glucose concentration, a dead spore is one that has no chance of germinating with the saturating concentration of glucose and, thus, no chance of germinating for lesser glucose concentration either. Hence measuring the percentage of spore bags that do not germinate after receiving a 2%-glucose tells us the percentage of spore bags that contain only dead spores.

We verified that all three drugs indeed inhibited GFP expression in yeast spores (Appendix Fig S21). Nearly 100% of the spores were still alive after we inhibited either ATP synthesis with antimycin A or global translation with cycloheximide for 24 h (Fig 4B). In contrast, inhibiting global transcription with thiolutin for 24 h killed almost all spores: less than 5% of the spore bags were still able to germinate with a 2%-glucose after the thiolutin treatment (Fig 4B). These results establish that transcription—and potentially RNA metabolism in general—is the major process that keeps dormant spores alive.

## Method devised for detecting RNA synthesis in dormant yeast spores at single-cell level

To further explore how transcribing during dormancy is connected to germination ability—and thus to survival—for a given glucose concentration, we first developed a method to measure all the RNA that is actively being made in dormant yeast spores with a single-cell resolution. We modified an existing method for mammalian cells in which the cells uptake 5-Ethynyl Uridine (5-EU) and incorporates it into RNA that is being synthesized, after which the cells are fixed and fluorophores (e.g. Alexa 488) enter the fixed cells to bind the 5-EU-labelled RNAs (Jao & Salic, 2008; Fig 4C). Since 5-EU would only be found in RNAs that are synthesized while the extracellular 5-EU was present, this method lets us determine the total amount of RNAs that are made during a specified time window (i.e. only while the 5-EU was present). The method therefore distinguishes these freshly made RNAs from the RNAs that were present before we added the 5-EU to the cell culture medium. By adapting this method for yeast, we could visually verify, by fluorescence, that yeast spores actively transcribe during dormancy and determine the amounts of RNA produced by the dormant spores, relative to one another, during a 24-h incubation in 5-EU (Fig 4D). To our knowledge, this is one of the first demonstrations of active transcription in dormant yeast spores with a single-cell resolution, as previous studies used either indirect or bulk assays that required lyses of many spores (Brengues *et al*, 2002). We used several methods to verify that our method worked in yeast spores (Appendix Figs S22 and S23).

## Total transcription rate in dormant spores is virtually uncorrelated with germination ability

To apply the adapted method to the GFP-inducible spores, we incubated the spores in PBS with both doxycycline and 5-EU for 24 h. Hence, the total RNA level accumulated in a day is a proxy for the total transcription rate of a dormant spore. We found a statistically significant but weak, positive correlation ($R = 0.24$) between the spores' total transcription rate (i.e. accumulated RNA level) and their GFP inducibility (Fig 4E). This, in turn, means that the total transcription rate poorly correlates with the spore's chance of germinating. Thus, the total transcription rate cannot explain why the GFP inducibility so precisely determines the germination ability for each glucose concentration. Yet, the fact that a 24-h exposure to thiolutin kills nearly all spores (Fig 4B) suggests that some aspect of transcription during dormancy—other than the total amount of transcription—should strongly correlate with the GFP inducibility and, in turn, with the germination ability for each glucose concentration. Since RNA polymerases I, II and III are required and central machineries for transcription, we next sought to address whether the amount of each RNA polymerase in a dormant spore strongly correlates with its GFP inducibility.

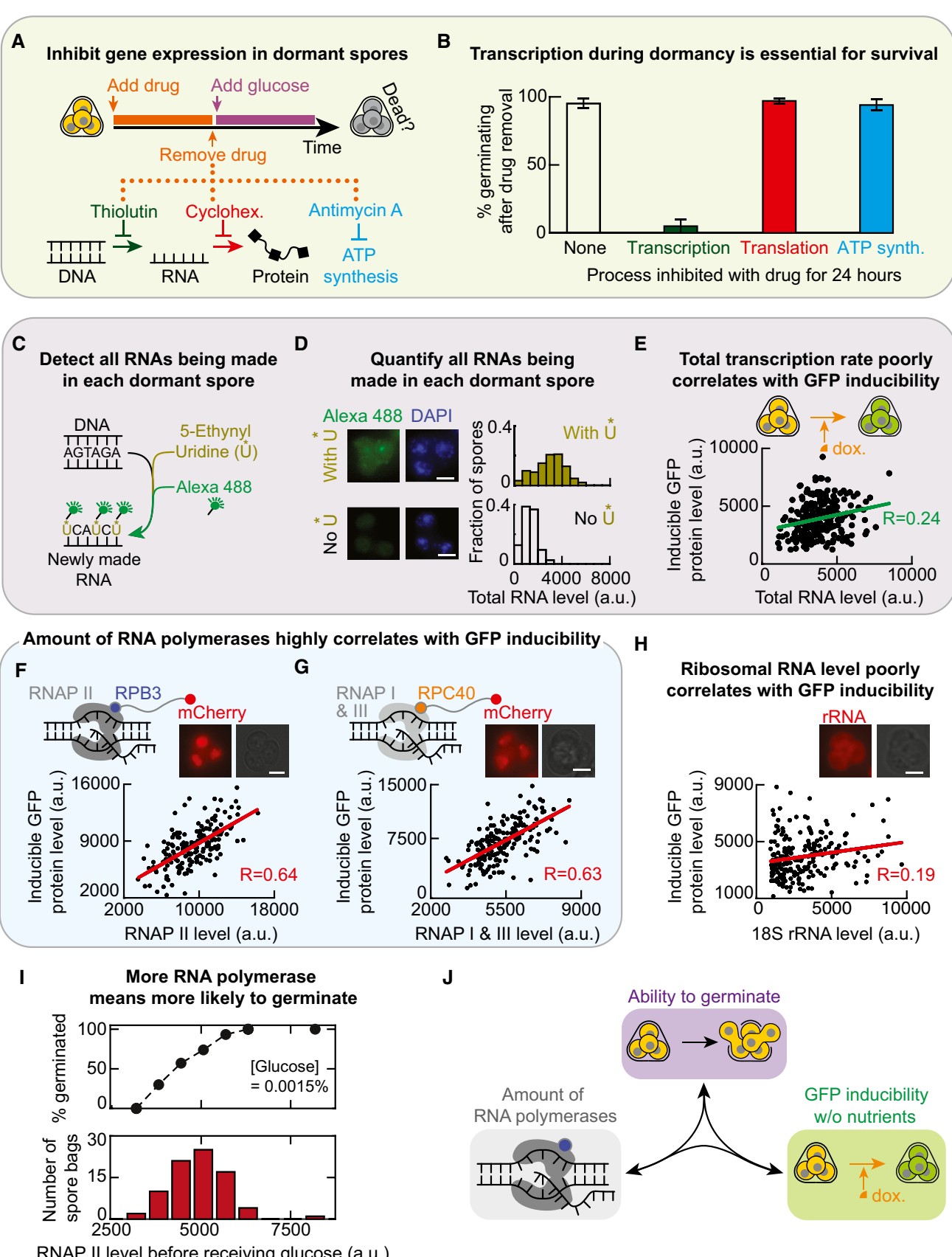

**Figure 4.**

**Figure 4.  More RNA polymerases I-III means higher germination ability and GFP inducibility.**

A  Protocol for (B). See "Protocol for Fig 4B" in Materials and Methods. Thiolutin inhibits transcription. Cycloheximide inhibits translation. Antimycin A inhibits ATP synthesis by inhibiting oxidative phosphorylation.

B  For experiment in (A), percentage of spore bags that germinated. $n = 3$; error bar are sem.

C  Method to detect all RNAs being made in yeast spores with a single-cell resolution. Spores were incubated for 24 h in PBS with 1 mM of 5-Ethynyl Uridine (5-EU; denoted U*) that incorporates into freshly made RNAs. We fix the spores afterwards and let fluorophore (Alexa 488) enter the spores and bind U* (see Materials and Methods). "Total RNA level" of a spore bag is the total fluorescence from all the 5-EU labelled RNAs.

D  Microscope images show a spore bag (from strain "TS3") after either incorporating 5-EU as described in (C) (top two images) or, as a control, following the protocol in (C) but without the 5-EU (bottom two images) (also see Appendix Fig S23A). Scale bar: 2 μm. Top histogram: total RNA level per spore bag with 5-EU (i.e. Alexa 488 fluorescence; $n = 103$ spore bags). Bottom histogram: fluorescence per spore bag in control population (i.e. without 5-EU; $n = 95$ spore bags). Also see Appendix Fig S22A.

E  Each dot is from a single spore bag ("TS3" strain) with GFP-inducing circuit (Fig 3A). For each spore bag, we measured its GFP protein level ("GFP inducibility") and total RNA level (5-EU fluorescence) after incubating the spores for 24 h in PBS with both 100-μg/ml doxycycline and 1 mM of 5-EU (see (C)). $n = 245$ spore bags. Alexa 594 fluorophore attached to 5-EU (see Materials and Methods). Green line: linear regression with $R = 0.24$ and Pearson $P$-value $= 0.00018$.

F  See "Protocol for Fig 4F" in Materials and Methods. GFP inducibility per spore bag (each dot, "TS8" strain) measured as in (E) but now with live time-lapse without the 5-EU. "RNAP II level" is the mCherry fluorescence per spore bag due to the mCherry protein fused to Rpb3, a subunit of RNA polymerase II. $n = 182$ spore bags; Red line: linear regression with $R = 0.64$ and Pearson $P$-value $= 3.02 \times 10^{-22}$. Scale bar = 2 μm.

G  Same protocol as in (F) but with "TS9" spores. "RNAP I & III level" is the mCherry fluorescence per spore bag due to the mCherry protein fused to Rpc40, a subunit of both RNAP I and RNAP III. $n = 185$ spore bags; Red line: linear regression with $R = 0.63$ and Pearson $P$-value $= 6.6 \times 10^{-22}$. Scale bar = 2 μm.

H  Same protocol as in (F) but with "TT14" spores fixed after 24 h of incubation in PBS with 100-μg/ml doxycycline. 18s rRNA level is from CAL Fluor Red 610 fluorescence emitted by single-molecule FISH probes bound to 18S rRNAs (see Materials and Methods). $n = 213$ spore bags; Red line: linear regression with $R = 0.19$ and Pearson $P$-value $= 0.005$. Scale bar = 2 μm.

I  Bottom: RNAP II levels of spore bags ("TS8" strain) in a population, measured as in (F). Top: As a function of the RNAP II level (binned in the histogram), percentage of spore bags that germinated after receiving a 0.0015%-glucose, averaged over all spore bags with the same binned RNAP II level. $n = 80$ spore bags.

J  Triangular relationship. Any pair of the following three are positively correlated: Germination ability for each glucose concentration (purple), GFP inducibility (green) and amounts of RNA polymerases I-III.

## Amount of RNA polymerases I-III are strong predictors of GFP inducibility and germination ability

To measure the amounts of RNA polymerase (RNAP) I, II and III in individual spores through live time-lapse microscopy, we genetically engineered two yeast strains. One strain produced Rpb3—a subunit of RNAP II—as a protein fused to the fluorescent protein, mCherry (Fig 4F). The other strain produced Rpc40—a subunit of both RNAP I and RNAP III—as a protein fused to mCherry (Fig 4G). Both strains also have the GFP-inducing synthetic circuit (Fig 3A). For these strains, the mCherry fluorescence was entirely confined to the nuclei of both spores and vegetative yeasts (Appendix Fig S24). Hence, the mCherry levels accurately reflected the amounts of each RNAPs in spores. After sporulating these two strains, we incubated the resulting spores in PBS with doxycycline for 24 h to induce their *GFP*. We then used a microscope to measure both the GFP and mCherry (RNAP) levels in individual spore bags. In one of the strains, mCherry level represents the level of RNAP II, which transcribes all the coding genes and also produces some non-coding RNAs. In the other strain, mCherry level represents the amount of RNAP I and RNAP III combined, both of which produce only non-coding RNAs such as ribosomal RNAs. Strikingly, the amount of RNAP II strongly ($R = 0.64$) and positively correlated with the GFP inducibility (Fig 4F). The combined amount of RNAPs I and III also strongly ($R = 0.63$) and positively correlated with GFP inducibility (Fig 4G). Hence, the amounts of both classes of RNA polymerases—one for transcribing mostly coding genes (RNAP II) and one for transcribing non-coding genes (RNAPs I and III)—are strong predictors of GFP inducibility. We also used single-molecule RNA FISH (Raj *et al*, 2008; Youk *et al*, 2010) to quantify the amount of 18S ribosomal RNA (18S rRNA)—a ribosomal subunit—in dormant spores that we fixed after the 24 h of doxycycline. We then determined how the amount of 18S rRNA in spores correlated with the GFP inducibility (Fig 4H). We found that the amount of 18S rRNA in spores barely

correlated with their GFP inducibility ($R = 0.19$). Since the amount of 18S rRNA is a proxy for the amount of ribosomes, this result suggests that the amounts of ribosomes in a dormant yeast spore is virtually uncorrelated—and at most weakly correlated—with GFP inducibility and thus with germination ability. This is consistent with transcription, rather than translation, being the dominant process for surviving dormancy (Fig 4B).

Since GFP inducibility strongly and positively correlates with the amounts of all three RNAPs and germination ability strongly and positively correlates with GFP inducibility, we would expect that the amount of RNAPs in a dormant spore should also strongly and positively correlate with the spore's germination ability as well. To verify this for RNAP II—we will focus on RNAP II from now on instead of RNAPs I and III because RNAP II transcribes all coding genes such as *GFP* as well as some non-coding genes—we took the spores with Rpb3 fused to mCherry and then incubated them in a low concentration of glucose (0.0015%). Then, using the same method that we used for measuring the germination landscape (Fig 3E), we determined how likely it is that a spore bag with a given amount of RNAP II germinates after receiving the 0.0015%-glucose. We found that spore bags with more RNAP II indeed were more likely to germinate (Fig 4I).

Taken together, the results so far establish a triangular relationship in which a strong, positive correlation exists between any two of the following three quantities in dormant yeast spores (Fig 4J): GFP inducibility without nutrients, amount of RNAPs (RNAP II level or level of RNAPs I and III combined), and the germination ability for each glucose concentration.

## Dormant spores gradually lose GFP inducibility as they age

With the triangular relationship, we now have a molecular view—through RNAP II level—and a functional view—through GFP inducibility—for studying how dormant yeast spores age,

which is what we originally set out to study. To age the dormant spores, we incubated the wild-type and GFP-inducible spores in either water or minimal medium without any glucose for days or weeks at 30°C. In this "ageing experiment", we found that the number of dormant (i.e. alive) spores—the ones that we could germinate when we gave them a 2%-glucose—decreased by similar rates over several weeks regardless of the strain type and regardless of whether we kept the spores in water or minimal medium (Fig 5A). Specifically, about half of the spores in the population died after ~30 days without glucose and almost everyone died after ~60 days without glucose (Fig 5A). Furthermore, we observed that spores needed more time to germinate if we kept them longer without nutrients (Appendix Fig S25), suggesting that germinating is becoming more difficult as the dormant spores age.

To measure the GFP inducibility in these ageing spores, we took some of the spores from the ageing population, incubated them with doxycycline in PBS for 24 h (Appendix Fig S26), measured the resulting GFP levels in these spore bags with a microscope (Fig 5B and Appendix Fig S27) and thus determined how the GFP inducibility changed over the 80 days in PBS without nutrients (Appendix Fig S27). Since we also observed dead spores in the ageing population, we could also measure the dead spores' GFP inducibility with doxycycline on different days. The GFP inducibility of dead spores remained at near zero (near background fluorescence value) regardless of when the spores had died (Fig 5B—grey points). The GFP inducibility of dormant spores, when averaged over every dormant spore bag, gradually decreased and approached the near-zero GFP inducibility of the dead spores that were in the same ageing population over tens of days (Fig 5B—blue points). The dormant spores' GFP inducibility eventually reached a barely detectable level—matching the dead spores' GFP inducibility—after ~80 days. By this time, nearly all spores had died (Appendix Fig S27). Together, these results show that dormant spores indeed lose their GFP inducibility over time as they age and approach their deaths. Crucially, in accordance with our revised hypothesis, we did not find any dead spores with a detectable GFP inducibility. Hence, dormant yeast spores lose their GFP inducibility *before* dying. Moreover, the fact that the oldest dormant spores have barely detectable GFP inducibility, just like the dead spores, suggest that GFP inducibility is nearly all lost moments before death.

## Dormant spores gradually have less copies of RNA polymerase II as they age

To further validate that dormant spores lose nearly all GFP inducibility just before dying, we sought to measure the RNAP II level in ageing spores since GFP inducibility strongly and positively correlates with the RNAP II level in spores. We repeated the ageing experiment but now with the spores that had the fluorescent RNAP II. By taking out aliquots of these spores from an ageing population and then using a microscope to measure the RNAP II levels in each of these aged spore bags, we found that spores' RNAP II level decreased as they aged. Moreover, after ~50 days of ageing, almost all the spores had barely detectable RNAP II levels which were near the background fluorescence value (Fig 5C and Appendix Figs S28 and S29).

## More RNA polymerase II means more likely to survive dormancy

After a month of ageing, more than ~50% of the spores were dead. This was the case, for example, on day 39 of ageing in water. We aliquoted a subset of the aged spores on day 39. Then, we used a microscope to measure the RNAP II levels in each of the aliquoted spore bags. Afterwards, we incubated these spores in minimal medium with a 2%-glucose and then, with time-lapse microscopy, determined which spore bags germinated (i.e. were alive) and which ones did not germinate (i.e. were dead). Spores that had more RNAP II had a higher chance of being alive than spores with less RNAP II (Fig 5D). In particular, we found that nearly 50% or more spores with the lowest observed RNAP II levels were dead. These spore bags had near background fluorescence level of ~2,000–3,000 fluorescence units (Fig 5D). Since GFP inducibility positively correlates with the RNAP II level, this finding complements the observation that spores lose almost completely their GFP inducibility just before dying, as also expected from the triangular relationship (Fig 4J).

## Aged spores at the edge of dormancy before dying have nearly depleted RNAP II and have strained gene-expressing ability

On the 39th day of ageing, many spores with barely detectable RNAP II levels (2,000–3,000 fluorescence units in Fig 5D) were dead but many others also with the same barely detectable RNAP II levels were alive, albeit with less than a 50% chance (Fig 5D). This 39-day-old population is thus ideal for understanding the last moments of dormancy—dormancy just before death. With this population, we sought to understand what actually distinguishes the dead from dormant spores when both have comparable, barely detectable RNAP II levels (i.e. 2,000–3,000 fluorescence units in Fig 5D). We examined two histograms of RNAP II levels: one for dead spores (Fig 5E—lower, grey histogram) and the other for dormant spores (Fig 5E—upper, blue histogram). The two histograms show many alive and dead spores having the nearly identical, low range of values for RNAP II level (Fig 5E). After measuring the RNAP II levels in individual spore bags, we incubated both the dead and dormant spore bags together in a minimal medium with a 2%-glucose and then tracked the RNAP II levels in individual spore bags with time-lapse microscopy. We found that, after receiving the 2%-glucose, the dead spores did not make any detectable levels of new RNAP II (Fig 5F—grey traces and points) whereas the dormant spores did make observable amounts of new RNAP II until they germinated (Fig 5F—blue traces and points). Moreover, these dormant spores often started to make new RNAP II shortly before buds appeared (i.e. before they germinated; Movies EV1–EV4, Fig 5F). They also, on average, took longer times to germinate (Appendix Fig S30). Strikingly, these aged dormant spores, near death, did not produce any detectable levels of new RNAP II for the first 5–10 h after receiving the 2%-glucose (Movies EV1–EV4, Appendix Fig S30). After this "lag time", they began producing new RNAP II and then germinated shortly afterwards.

These results paint a picture in which dormant spores with barely detectable amounts of RNAP II are struggling through gene expression and germination after receiving a 2%-glucose, as expected intuitively from dormant spores that are near death. This picture also makes sense given that RNA polymerases, including

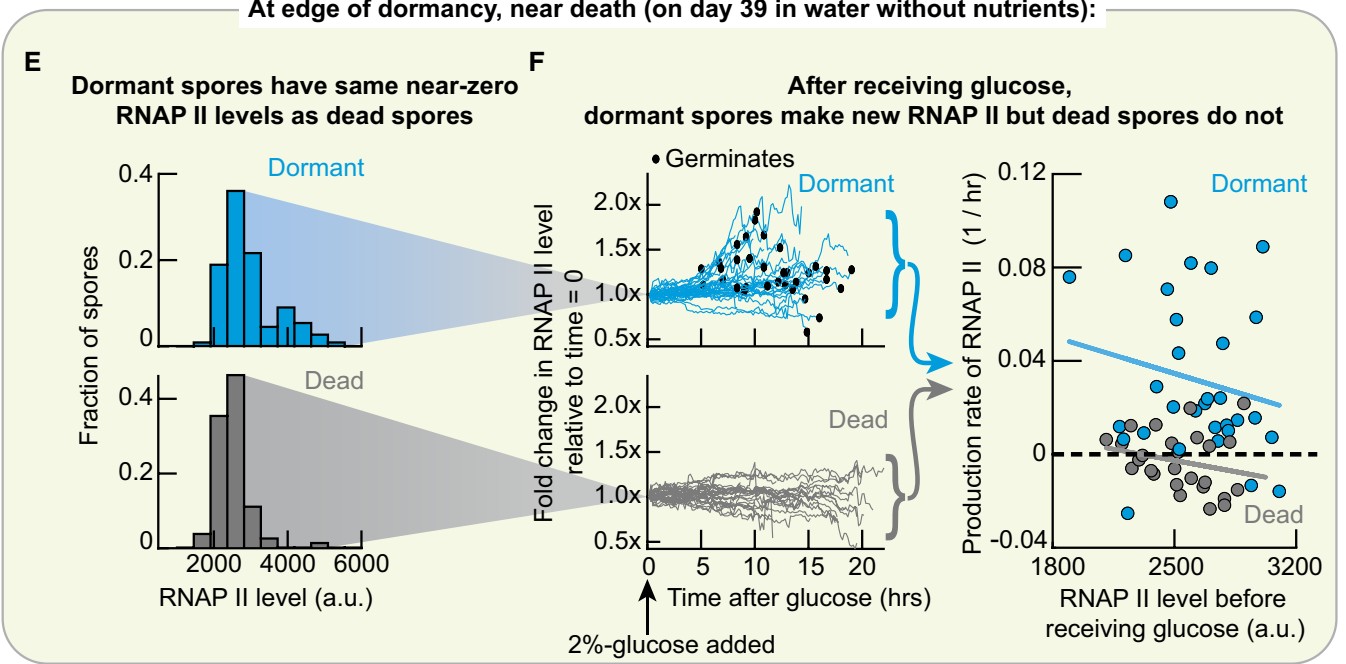

**Figure 5.**

---

**Figure 5.   Dormant spores gradually have less RNA polymerase II and GFP inducibility as they age, eventually losing all gene-expressing ability just before dying.**

A   Percentage of wild-type and GFP-inducible spore bags (Fig 3A) that germinate due to a saturating glucose concentration (2%) that they encounter after ageing for 0–85 days in water (grey and black) or minimal medium with essential amino acids (red) at 30°C. $n = 3$; error bars are sem.

B   Mean GFP inducibility of dormant (blue) and dead (grey) spore bags that were incubated in water without nutrients at 30°C over ~80 days (also see Appendix Figs S27 and S28). Blue data points are the average GFP levels from the histograms shown in Appendix Fig S27. Grey data points are averages of the dead spore bags' histograms shown in Appendix Fig S27. $n = 3$, error bars show the average standard deviation from three biological replicates.

C   Average RNAP II level of spore bags incubated in water for 50 days at 30°C. $n = 3$, error bars represent the average standard deviation in a population, each population having at least 100 spore bags (see Appendix Figs S28 and S29).

D   Top: Percentage of spore bags, having the same RNAP II level on 39th day of ageing in water (as in (A)), germinating after receiving a 2%-glucose. Percentages are averaged over all spore bags with the same binned RNAP II level (corresponding histogram shown below). $n = 193$ spore bags.

E   RNAP II levels of dormant (alive) spore bags in blue and of dead spores in grey; both from the same population on 39th day of ageing in water (as in (A)) before any glucose added. $n = 111$ spore bags (blue); $n = 82$ spore bags (grey).

F   Corresponds to histograms in (E). Blue is for dormant and grey is for dead spore bags. Line traces show, after receiving a 2%-glucose, the fold change in each spore bag's RNAP II level relative to the spore bag's initial RNAP II level (i.e. value just before we added the 2%-glucose). Black dots represent moments of germination. $n = 38$ for blue and $n = 32$ for grey. Data points in the rightmost plot show the RNAP II production rate in each spore bag profiled in the line traces. For each spore bag, we fitted an exponential function to its RNAP II level over time (one of the line traces). Average production rate of RNAP II in dormant spore bags (average over all blue points) is a positive value, 0.031/h and in dead spore bag (average over all grey points) is a negative value, −0.0030/h. For a *t*-test on the difference between the two values: *P*-value $= 1.4 \times 10^{-5}$. Linear regression line for dormant (blue) is R = −0.19 and for dead (grey) is R = −0.23.

RNAP II, are required to make more RNAPs I-III and that the aged, nearly dead spores have barely detectable amounts of RNAP II. Crucially, these results establish that dead spores, which have virtually undetectable GFP inducibilities during dormancy (Fig 5B), also have virtually undetectable gene-expressing abilities after they receive a saturating amount (2%) of glucose. This is supported by the fact that RNAP II—required for expressing all coding and some non-coding genes—is at barely detectable levels in the aged spores that are near death (Fig 5E) and that a 2%-glucose does not induce gene expressions that lead to the making of a new RNAP II (Fig 5F).

**Transiently inhibiting transcription during dormancy permanently decreases GFP inducibility**

We found that ageing spores gradually lose the GFP inducibility. In other words, older dormant spores, when forced by doxycycline to express *GFP* without nutrients, yield less steady-state GFP levels. Although dormant spores have less RNAP II as they age (Fig 5C), it is not obvious that less RNAP II should lead to less steady-state GFP level after 24 h of doxycycline. This is all the more since we showed above that the GFP level reaches a steady-state value because the spores stop producing GFP, despite the doxycycline not degrading and still being present at the saturating concentration at the end of the 24-h incubation (Appendix Figs S13 and S14). To understand why spores lose the GFP inducibility as they age, we sought a clearer understanding of gene expression in general, which occurs during dormancy as we have shown with the 5-EU labelling of RNAs (Fig 4E).

As a start, we incubated the GFP-inducible spores in water with thiolutin for various hours (between 0 and 24 h). Afterwards, we washed away the thiolutin and then added doxycycline for 24 h to induce their GFP expression (Fig 6A). At the end of the 24-h incubation with doxycycline, we measured the GFP inducibility of each spore bag. After measuring the GFP inducibility, we incubated the spores in a 2%-glucose to determine which ones were dormant (alive) and which were dead. In this experiment, we found that the GFP inducibility of dead spore bags always remained at the minimally observable values, as expected (Fig 6B—grey points). The GFP inducibility of dormant spore bags gradually decreased as the

hours spent in thiolutin increased, until the GFP inducibility reached barely detectable levels after ~15 h in thiolutin (Fig 6B—blue points). Hence, removing thiolutin did not restore the spore's GFP inducibility back to the level that it had before it encountered thiolutin. In other words, transiently inhibiting global transcription for ~12 h or more has permanently (irreversibly) reduced the GFP inducibility. Along with this irreversible decrease in GFP inducibility, we observed that thiolutin accelerated deaths, resulting in almost all spores being dead after ~24 h with thiolutin as opposed to the ~2 months in the previous ageing experiment without thiolutin (Fig 5A and Appendix Fig S31A). These results establish that if dormant spores do not produce transcripts—and potentially proteins from these transcripts—during a day, then they irreversibly and completely lose GFP inducibility and die within a day. Thus transcribing—and potentially translating—during dormancy during a day is required for surviving and directly determines how much GFP the spore can make when it is forced to do so.

**Spores that died due to transient inhibition of transcription do not express genes despite having abundant RNAP II**

To better understand how the transcription inhibition by thiolutin causes death, we focused on spores that encountered thiolutin for 12 h since many of them were either dead or nearly dead, thereby affording us enough dead and nearly dead spores to compare. During the 12 h of thiolutin, the spores' production of RNAP II itself would be stopped. Indeed, these spores had ~30% less RNAP II after the thiolutin treatment than they had before the treatment. They also had ~30% less RNAP II after the thiolutin treatment than the spores that we kept in water for 12 h without thiolutin (Appendix Fig S31). But these thiolutin-treated spores still had abundant RNAP IIs, nearly 2–3 times more than 39-day-old spores that we kept in water without thiolutin (Fig 6C). Surprisingly, however, ~50% of the spores were dead after the 12-h of thiolutin, which we discovered by incubating the spores in a 2%-glucose after washing away the thiolutin (Appendix Figs S31A and 6D). In comparison, without thiolutin, we would need to wait ~39 days for ~50% of the ageing spores to be dead (Appendix Fig S32).

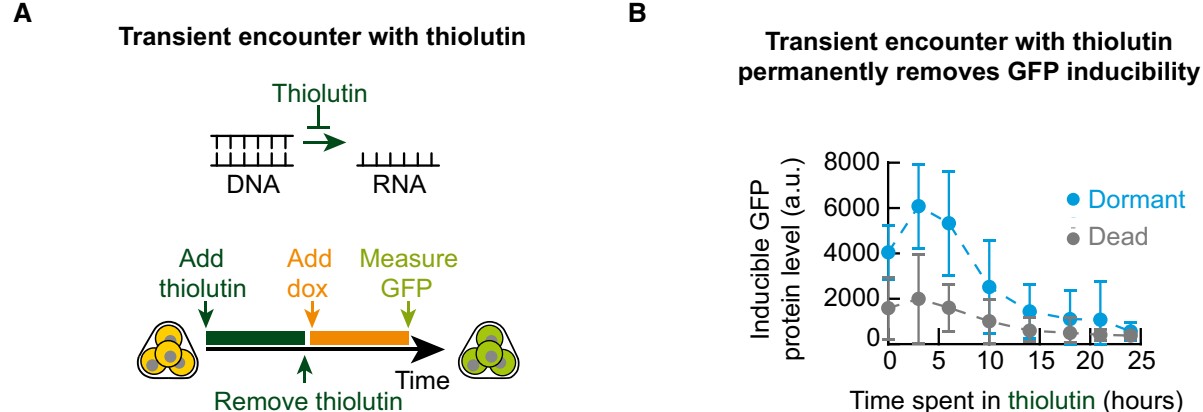

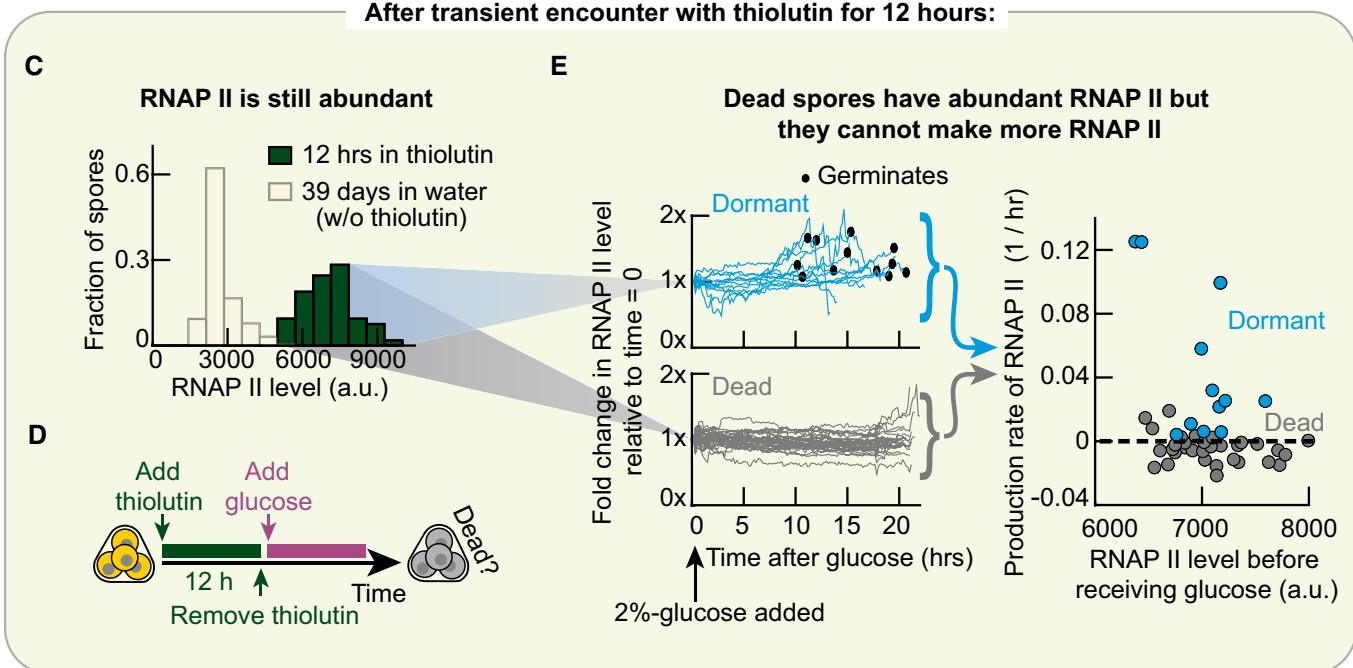

**Figure 6. Stopping transcription for a day causes dormant spores with abundant RNA polymerases to lose all gene-expressing ability and thus die.**

A   Protocol for (B-C) (See "Protocol for Figs. 6A–C" in Materials and Methods).

B   For protocol in (A), mean GFP inducibility of dormant (blue) and dead (grey) spore bags as a function of amount of time spent in thiolutin. $n = 3$, error bars represent the standard deviation among spore bags in a population, averaged over three populations.

C   RNAP II level of spore bags after 12 h of thiolutin (green; protocol in (A)) or after 39 days of ageing in water without drugs (light yellow; data from Fig 5C).

D   Protocol for (E) (See "Protocol for Fig 6E" in Materials and Methods).

E   Corresponds to green histogram in (C). Blue is for dormant and grey is for dead spore bags. Line traces show, after receiving a 2%-glucose, the fold change in each spore bag's RNAP II level relative to the spore bag's initial RNAP II level (i.e. value just before we added the 2%-glucose). Black dots represent moments of germination. $n = 12$ for blue and $n = 42$ for grey. Data points in the rightmost plot show the RNAP II production rate in each spore bag profiled in the line traces. For each spore bag, we fitted an exponential function to its RNAP II level over time (one of the line traces). Average RNAP II production rate in dormant spore bag (average over all blue points) is a positive value, 0.044/h and in dead spore bag (average over all grey points) is a negative value, −0.0045/h. For a $t$-test on the difference between the two values: $P$-value = $6.5 \times 10^{-3}$.

To test whether the RNAP II-abundant spores that died during the 12 h of thiolutin could express genes after receiving a 2%-glucose, we used time-lapse microscopy to track the RNAP II levels in both dead and dormant spores after washing away the thiolutin and then incubating the spores in minimal medium with a 2%-glucose (Fig 6E—left half). As with the spores that aged for 39 days without thiolutin (Fig 5F), the thiolutin-treated dormant spores made new RNAP II after receiving the 2%-glucose, typically taking ~5–10 h before producing detectable amounts of new RNAP II (Fig 6E—blue curves and points) just like the 39-day-old spores

with little RNAP II. As for the spores that died from the thiolutin treatment, we did not observe any of them producing noticeable amounts of new RNAP II during the ~20 h of incubation in the 2%-glucose, despite all of them having abundant RNAP IIs (Fig 6E—grey curves and data).

Taken together, the thiolutin experiments (Fig 6B–E) establish that dead spores can have many copies of RNAP II and yet a 2%-glucose may not induce gene expression in them, including those necessary for producing RNAP II. These results support the idea that spores need to produce some transcripts, within several hours (~12 h as a conservative estimate), in order to maintain GFP inducibility—and thus for a 2%-glucose to be able to induce gene expression in them—and hence for the spores to survive. Moreover, the fact that transiently inhibiting transcription permanently reduces GFP inducibility (Fig 6B) strongly suggests that during the transcription inhibition, spores lost copies of key molecules that are required for gene expression and that the spores could not fully replenish these molecules after transcription became permissible again. In other words, the experiments suggest that the production rates for these molecules is lower than their degradation rates during dormancy. This is also consistent with the fact that dormant yeast spores eventually die if left they are left alone in water without any drugs (Fig 5A)—there is a net loss of the molecules because, during dormancy, the production rates for the molecules are lower than the degradation rates for the molecules.

## Globally inhibiting gene expression shortens spores' lifespans

To end our study, we tested the idea proposed above by estimating the production and degradation rates of key molecules that are necessary for gene expression during dormancy. To achieve this at a systems-level—without having to identify all such molecules and then measure their production and degradation rates—we continuously inhibited transcription and translation while ageing the spores that have the fluorescent RNAP II for ~2 months in water with either a saturating concentration of cycloheximide or thiolutin. On various days, we took out an aliquot of spores from the ageing population, washed away the drug from the aliquoted population, measured the RNAP II level in these spores, and then incubated them in minimal medium with a 2%-glucose to check how many were still alive (i.e. dormant; Fig 7A). From these measurements, we determined the rate at which a spore's ability to express genes during dormancy decreases—due to the net loss of key molecules required for gene expression—by measuring the net loss rate of RNAP II as a proxy (Fig 7B and C). The RNAP II level averaged over all dormant spores decreased by half after ~8.4 days of ageing in cycloheximide (translation inhibitor; Fig 7B and C—red bar). As a comparison, the RNAP II level averaged over all dormant spores decreased by half after ~14 days if the spores were left to age in water without any drugs (Fig 7C—white bar). Hence, inhibiting translation accelerates the net loss of RNAP II by ~1.5 times compared with natural (drug-free) dormancy. Moreover, with cycloheximide, 50% of the spore population were dead after ~19 days (Fig 7D and E—red bar) whereas, without any drugs, half of the spore population were dead after ~38 days (Fig 7E—white bar). The approximately 2-fold difference caused by cycloheximide in both the rate of RNAP II loss and the half-life of dormancy supports the idea that dormant spores age and eventually die because they gradually

lose enough of the molecules that are required for gene expression, including RNAP II, and that spores cannot replenish these after receiving a 2%-glucose.

The fact that cycloheximide accelerates the net loss of RNAP II proves that RNAP II production, which requires transcription and translation, occurs during dormancy. Cycloheximide prevents spores from producing RNAP II and thus accelerates the RNAP II loss. By how much it is accelerated tells us the production rate of RNAP II during dormancy. We found that dormant spores in water without any drugs produced RNAP II at an extremely slow pace, having a characteristic production time of ~30.3 days (Appendix Fig S33). In comparison, after receiving a 2%-glucose but before germinating, spores produced new RNAP II 150–760 times faster than they did during dormancy (i.e. before receiving the 2%-glucose; Fig 5E and Appendix Fig S33). The extremely slow production of RNAP II during dormancy supports the idea that spores do not replenish the molecules required for production, such as RNAP II, fast enough for preventing their net losses. As RNAP II is certainly required for expressing coding genes and thus for germination (i.e. for building a new cell after receiving a 2%-glucose), the net loss of RNAP II ensures that dormant spores will eventually die if left alone. Our measurements on RNAP II also strongly suggest that the molecules required for gene expression, which are lost on the order of at most several hours during the thiolutin treatment (Fig 6B), require transcription for them to be made and are relatively unstable compared with macromolecular machineries such as RNAP II that are also required for gene expression. In fact, since the RNAP II fluorescence arises from a single protein, Rpb3, fused to mCherry, the slow degradation of RNAP II also means that smaller proteins likely also degrade over days, not hours, during dormancy. Taken together, these results suggest that key transcripts, which are required for gene expression, are lost on the order of at most several hours during the transcription inhibition. In accordance with this view, spores retained more than half of their original RNAP II level after a day thiolutin (Fig 7C) yet the number of surviving spores were halved after just ~12 h (Fig 7E).

## Model for dormancy-to-death transition

Taken together, our experiments suggest a model of how dormant spores gradually approach their deaths (Fig 7F). In this model, when yeast spores enter dormancy sometime after sporulation (Brengues *et al*, 2002; Thacker *et al*, 2011), they initially possess high amounts of key molecules that are necessary for gene expression which include, but are not limited to, RNA polymerases. Spore bags would vary in the initial amounts of these key molecules—leading to different germination abilities among them for each glucose—but all of them initially have sufficiently high amounts of the molecules to let them germinate to a 2%-glucose. During dormancy, retaining the ability to express genes is crucial for being able to germinate and thus for survival because a spore must build a new cell after glucose appears (and thus express genes that code for the building blocks of the new cell). The molecules required for gene expression become less abundant as time passes without any nutrients because spores cannot replenish these molecules faster than the rate at which they degrade. The resulting net loss of these molecules over time leads to the spores gradually losing the ability to express genes such as those involved in making RNAP II. The

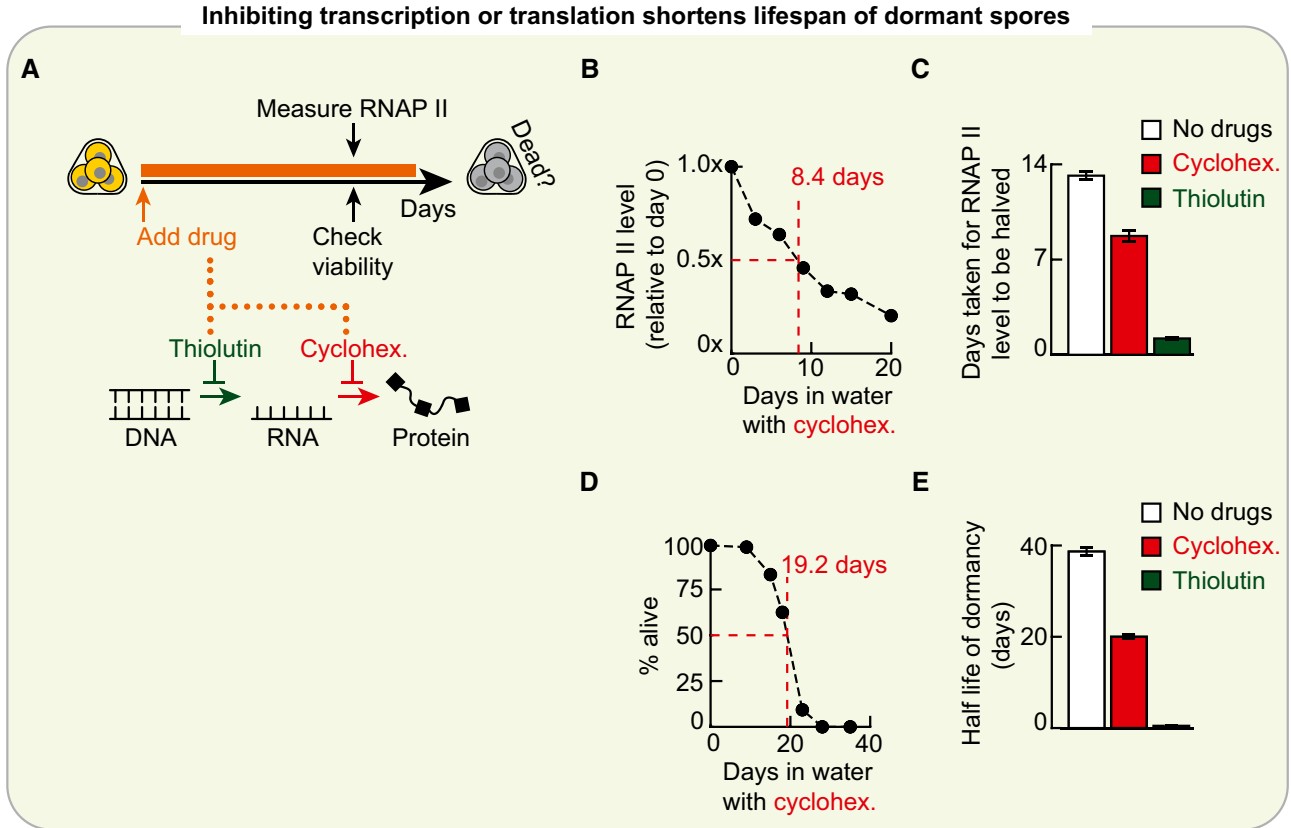

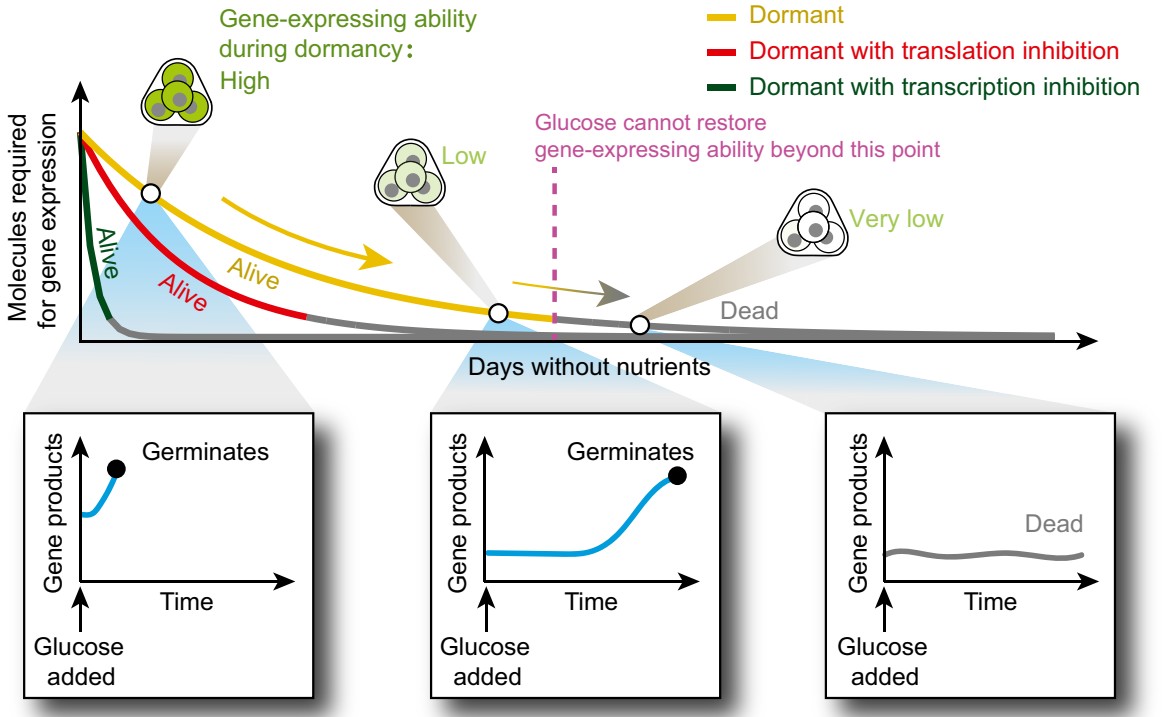

**Figure 7.**

Figure 7. **Gradual loss of molecules that are required for gene expression underlies ageing and eventual death of dormant yeast spores.**

A  Protocol for (B–E). See "Protocol for Fig 7B–E" in Materials and Methods. Two drugs: Thiolutin inhibits transcription and cycloheximide inhibits translation.
B  Experiment in (A) with cycloheximide. For one representative population with cycloheximide (other replicates and data for thiolutin are in Appendix Fig S32). RNAP II level relative to its day 0 value, averaged over all dormant spore bags in a population. We used a linear interpolation determine the time taken for the halving of RNAP II level (8.4 days).
C  Experiment in (A). Time taken for the RNAP II level to decrease by half in dormant spores without drugs (white), with cycloheximide (red) and with thiolutin (green). For spores with thiolutin, we extrapolated this value from the measurements because, even when every spore had died after 2 days of thiolutin, average RNAP II level was still more than half its initial level (Appendix Fig S31). $n = 3$; error bars are sem.
D  Experiment in (A). For one representative population with cycloheximide (other replicates and data for thiolutin are in Appendix Fig S32). Time taken for half of the spore population to die ("half-life of dormancy"), estimated by linear interpolation (19.2 days).
E  Experiment in (A). Time taken for half of the spore population to die for populations without drugs (white), with cycloheximide (red), and with thiolutin (green). $n = 3$; error bars are sem.
F  Putting all our data together leads to a conceptual model of how spores age during dormancy and eventually die: key molecules required for gene expression, including RNA polymerases, are gradually lost during dormancy because spores cannot produce them faster than they degrade. These molecules eventually reach below some threshold amounts after which the spore has permanently lost the ability to express genes, meaning that even ample glucose cannot restore the gene-expressing ability (inset at bottom right). Since, after receiving glucose, spores must express genes in order to build a new cell that will bud off the spore (i.e. germinate), the permanent loss of gene-expressing ability at the final moments of dormancy marks the spore's death (grey parts of the curves in the top plot).

spores also become less able to germinate as days pass by, requiring more glucose to guarantee that they germinate as they age (Fig 3G). An aged spore that has lost the gene-expressing ability can neither build a new cell when a 2%-glucose appears—so it cannot germinate—nor can it make the lost molecules that are crucial for regaining the gene-expression ability when a 2%-glucose appears. Accordingly, dead spores cannot express genes including *GFP* (Fig 5B) and those involved in the making of RNAP II when a 2%-glucose appears (Fig 5F). Once the amounts of the molecules required for gene expression decrease below certain threshold values—note that different molecules can have different threshold values—the spore cannot germinate even with a saturating amount of glucose and is thus considered dead. Our results suggest that among the many molecules that are required for gene expression, the key molecules—the ones that degrade the fastest during dormancy—are RNA polymerases and, likely, certain crucial transcripts which may be non-coding transcripts. Further studies, however, are required to definitively show that such transcripts play a role and identify them.

## Discussion

Our study began by posing two broad, related questions: how an organism whose life has almost ceased—including dormant organisms—remain alive and how we can monitor its approach to death given that it has faint signatures of intracellular dynamics that are difficult to discern. We addressed key aspects of these two questions in the case of dormant yeast spores while also raising new questions that invite future investigations. A conceptual difficulty in addressing these questions lies in the phrase, "almost ceased". Prior to our work, how slowly intracellular processes such as gene expression occur in dormant yeast spores has been unclear as firm numbers were missing. With single-cell level measurements, we identified key quantities that are relevant for studying ageing during dormancy: probability of germinating (which we called "germination ability"), the ability to express genes during dormancy when forced to so (proxied by the "GFP inducibility"), and the amount of RNAP polymerases inside the spores (Figs 1–4). After identifying these, we put firm numbers on them and showed how their decays

over time collectively represent the process of ageing during dormancy for yeast spores (Figs 5–7). We found that the RNAP II production and degradation take days whereas they take minutes in vegetative yeasts. We rigorously quantified the germination ability for every glucose concentration and found that it decays over days. With the GFP-inducing synthetic circuit, we quantified the ability of a dormant spore to express genes without nutrients when forced to do so and found that it also decays over days. Finally, we measured the lifespan of dormancy and found that one can shorten it from months—for spores kept in water without any drugs at 30°C—to either hours (by inhibiting transcription during dormancy) or days (by inhibiting translation during dormancy). We revealed, with a single-cell resolution, how these quantities decayed over time up to the last moments of dormancy (i.e. just before the yeast spores die) and then determined when a yeast spore is no longer dormant but dead. Crucially, we found that these quantities are linked to one another and therefore we can study the dormancy-to-death transition by monitoring several different quantities. Of these, the key relationship that we uncovered is the ability to express genes being directly linked to the ability to germinate: a yeast spore cannot germinate when glucose appears after it has sufficiently lost the ability to express genes. After all, a yeast spore would need to express genes to build a new cell that will bud off the spore (i.e. germinate). By linking the different quantities and measuring the extremely slow production rates—compared with degradation rates—of gene expression machineries such as RNAP II, we showed that a complete loss of gene-expressing ability is inevitable for a yeast spore. These measurements led us to a new conceptual model for how yeast spores gradually lose dormancy (Fig 7F). One way to monitor the gradual loss of dormancy is by measuring the gradually decreasing ability of the yeast spore to express a generic gene when forced to do so without nutrients (e.g. GFP inducibility). For this reason, we propose here a concept of "amount of dormancy". The amount of dormancy gradually decreases in a yeast spore and once it goes below a certain amount, the spore dies because it cannot express genes even when glucose appears.

There are several proxies for the amount of dormancy, such as the GFP inducibility and the amount of RNAP II. But one class of molecules, which we did not directly identify and measure here, are key transcripts that we argue are likely important for maintaining

gene-expressing ability in yeast spores. While RNA polymerases are, as expected, required and important for maintaining gene-expression ability during dormancy, we have shown that ample RNA polymerases and ribosomes themselves are insufficient for yeast spores to maintain gene-expressing ability during dormancy. In fact, we found that after several hours of transcription inhibition, yeast spores with ample RNAP II and ribosomes cannot express coding genes both during dormancy and after receiving a saturating concentration of glucose—these spores died. By comparing the time-scale of protein degradations in spores with the timescale of deaths that occur during the transcription inhibitions, we argue that certain transcripts—perhaps non-coding transcripts—that degrade on a timescale of at most hours are required for maintaining gene-expression ability and, in turn, for keeping spores alive. We expect that future studies will investigate this interpretation. Such future studies may benefit from the method that we have, for the first time, adapted for yeast spores here: 5-Ethynyl Uridine (5-EU) labelling of new RNAs being made during dormancy.

On a related topic, our work motivates an avenue of research that is underexplored: understanding the central dogma of molecular biology in eukaryotic spores by using dormant yeast spore as a model system. With a synthetic gene-circuit, we showed that inducing transcription and translation of a generic gene is possible in dormant yeast spores without any external energy source. We provided, to our knowledge, one of the first evidences of active gene expression inside intact, dormant yeast spores at a single-cell resolution. This approach—building synthetic circuits in dormant yeast spores—may enable us to reveal fundamental principles of eukaryotic transcription and translation in dormancy. As a starting point, our work raises a fundamental question: how does a dormant yeast spore globally repress gene expression given that its total transcription rate appears to be nearly uncorrelated with the amount of RNA polymerases that it has? We believe that a good starting point for addressing this question is our finding that *GFP* expression stops after a day because the production—either transcription or translation—stops despite the inducer (doxycycline) still being present. Moreover, we may gain additional insights into how the central dogma of molecular biology functions in dormant eukaryotic spores by further investigating the glucose-triggered gene expressions in the primed, dormant yeast spores (Fig 2G).

While much is known about bacterial spores, particularly *B. subtilis* spores (Ablett *et al*, 1999; Suel *et al*, 2007; Dworkin & Shah, 2010; Segev *et al*, 2012; Parry *et al*, 2014; Sinai *et al*, 2015), the dormant state of yeast spores—a model for eukaryotic spores—has remained comparatively underexplored. Our discoveries on yeast spores may not apply to bacterial spores such as *B. subtilis* spores and determining the similarities and differences between dormant state of yeast spores with dormant states of bacterial spores is challenging because dormancy of both remain incompletely understood. To bridge the gap, one may adapt the approach that we introduced here: building synthetic gene circuits in bacterial spores (e.g. *B. subtilis* spores) and then studying their dormancy-to-death transition. Since eukaryotic gene regulations have many features that are distinct from those of prokaryotes, we expect that studies of gene regulation in dormant bacterial spores—another underexplored topic—along with further studies of gene regulation in dormant yeast spores will deepen our understanding of both forms of

dormancy and help us elucidate the similarities and difference between the two.

We hope that our study motivates future studies that find systems-level metrics for monitoring how dormancy in a cell or multicellular organism gradually transitions to death. Collectively, such studies will likely deepen our understanding of dormancy-to-death transitions.

# Materials and Methods

### Protocol for Fig 4B

We first incubated spores (strain "TT14") for 24 h in PBS with either 10 μg/ml of thiolutin (inhibiting transcription) or 200 μg/ml of cycloheximide (inhibiting translation), or 100 μg/ml of antimycin A (inhibiting ATP production by inhibiting oxidative phosphorylation), or without any drugs. Afterwards, spores were washed four times with PBS and then incubated in minimal medium with 2% glucose for 24 h.

### Protocol for Fig 4F

We simultaneously measured the GFP inducibility and RNAP II level in individual spore bags after 24 h of incubation in PBS with 100-μg/ml doxycycline (also see "Microscope Data Analysis" section). "RNAP II level" is the mCherry fluorescence per spore bag due to the mCherry protein fused to Rpb3, a subunit of RNA polymerase II.

### Protocol for Figs. 6(A–C)

We incubated spores for different amounts of time in PBS with 10 μg/ml of thiolutin. Then, after washing away the thiolutin four times with PBS, we incubated the spores for 24 h in PBS with 100 μg/ml of doxycycline. We then measured the GFP levels ("GFP inducibility") of each spore bag on a microscope. Then, with the spores still under observation in the microscope, we washed away the doxycycline three times with PBS and then incubated the spores for 24 h in minimal medium with a 2%-glucose to assign a state (dormant or dead) to each spore bag.

### Protocol for Fig 6E

After incubating spores for 12 h with 10 μg/ml of thiolutin, we washed away the thiolutin four times with PBS. We then place the spores under a microscope and incubated them for 24 h in minimal medium with a 2%-glucose. Spore bags that germinated within the 24 h of glucose were counted as dormant (alive). The ones that did not germinate during this time were counted as dead.

### Protocol for Fig 7(B–E)

We incubated a population of spores ("TS8" strain) in water with either 10 μg/ml of thiolutin or 200 μg/ml of cycloheximide, or without any drugs. At various time points during the incubation, we aliquoted some of the spores from the population and then, with a microscope, measured the mCherry fluorescence of each spore bag (representing RNAP II level since these spores had mCherry protein

fused to Rpb3, a subunit of RNAP II). The "RNAP II level" for each spore bag is the maximal mCherry fluorescence obtained from 36 z-stacks with a z-step 0.2 μm. After measuring the RNAP II level, we kept these spores under the microscope and washed away the drug four times with PBS. Then, while still keeping the spore bags under the microscope, we incubated them for 24 h in minimal medium with 2%-glucose. We measured the percentage of spore bags that germinated during the 24 h (these are counted as dormant (alive)).

## Drugs used in Fig 4A

We used thiolutin from Sigma-Aldrich (CAS Number 87-11-6). For storage, we aliquoted thiolutin into DMSO at a final concentration of 1 mM. We used cycloheximide from Sigma-Aldrich (CAS Number 66-81-9), which we maintained in DMSO at a final concentration of 100 mg/ml. We used antimycin A from Sigma-Aldrich (CAS Number 1397-94-0), which we diluted into ethanol at 10 mg/ml.

## Strains

The "wild-type", homozygous diploid yeast strain that we used is from Euroscarf with the official strain name "20000D" and genotype as follows: *MATa/MATα his3-11_15/his3-11_15; leu2-3_112/leu2-3_112; ura3-1/ura3-1; trp1Δ2/ trp1Δ2; ade2-1/ade2-1; can1-100/can1-100*. This strain generated four genetically identical, haploid spores for each spore bag. For engineering the GFP-inducible spores, we started from the haploid versions of "20000D", which were also from Euroscarf. These haploid strains were "20000A" (isogenic to another standard laboratory strain called "W303" with mating type "a") and "20000B" (isogenic to W303 with mating type "alpha"). The 20000A's genotype is as follows: *MATa his3-11_15; leu2-3_112; ura3-1; trp1Δ2; ade2-1; can1-100.* The 20000B's genotype is exactly the same as 20000A's, except that it is of the opposite mating type (mating type "alpha"). The GFP-expressing diploid strain, called "TT14", is nearly identical to the wild-type's except for the addition of selection-marker genes that we introduced during the construction of the strain. TT14 has the following genotype: (*MATa/MATα his3-11_15/his3-11_15; leu2-3_112/leu2-3_112; ura3-1/ura3-1; ADE2/ADE2; can1-100/can1-100; HygB/HygB; trp1Δ2/TRP1; URA3/ura3-1; pADH1-rtTA/pADH1-rtTA; pTET07-GFP/pTET07-GFP*). Here, *pADH1* is the constitutive promoter for *ADH1* gene in yeast (631 bases upstream of *ADH1*'s ORF), *rtTA* is the reverse tetracycline-controlled transactivator whose transcription-activation domain is from the yeast's Msn2, and *pTET07* is the promoter with seven binding sites for rtTA. We sporulated TT14 to form the "GFP-inducible spores" (Fig 3A). These constructs are more fully described in a previous publication (Youk & Lim, 2014). For experiments involving the metabolic labelling of RNA with 5-Ethynyl-Uridine (5-EU; Fig 4C), we had to obtain spores that were able uptake the extracellular 5-EU for incorporation into intracellular RNA. To achieve this, we engineered a diploid yeast strain, "TS3", which is the same as TT14—TS3 also has the same GFP-inducing synthetic circuit (Fig 3A) as TT14—but now with additional genotype modifications (*HIS3/HIS3; hENT1-ADH1/hENT1-ADH1; HSV-TKGDP/HSV-TKGDP*) for which we used a construct that was previously shown to let yeast import BrdU (Viggiani & Aparicio, 2006). *hENT1* is the "human Equilibrative Nucleoside Transporter", under the control of the promotor for *ADH1, HSV* is the Herpes Simplex Virus thymidine kinase under control of the promotor for *GPD1*, and *HIS3* is a selection marker gene used for the cloning. To obtain the spores whose subunits of RNA polymerases were fused to mCherry (Rpb3 for RNAP II and Rpc40 for RNAP I and III; Figs. 4F and 4G), we engineered two diploid strains, "TS8" and "TS9", which were the same as "TS3" —TS8 and TS9 both have the GFP-inducing synthetic circuit and allow 5-EU uptake)—but now with additional genotype modifications: (*RPB3-mCherry-NatR/ RPB3-mCherry-NatR*) or (*RPC40-mCherry-NatR/ RPC40-mCherry-NatR*). Here "*RPB3-mCherry-NatR*" is the *RPB3* gene on the yeast chromosome IX (YIL021W) fused at its 3'-end to the *mCherry* gene, using the protein-linker sequence from (Sheff & Thorn, 2004). *NatR* is the Nourseothricin resistance marker inserted behind the stop codon of the *RPB3-mCherry* fusion gene. "*RPC40-mCherry-NatR*" is the same as the "*RPB3-mCherry-NatR*" but now with the *RPC40* gene that is on the yeast chromosome XVI (YPR110C) instead of *RPB3*.

## Strain construction

For each yeast transformation, we integrated a single copy of an appropriate, linearized yeast-integrating plasmid at a desired genomic locus through a homologous recombination. We first introduced a promoter of *ADH1* controlling rtTA expression *pADH1-rtTA* into 20000A (wild-type haploid, mating type "a") and 20000B (wild-type haploid, mating type "alpha") at the HO locus, by inserting a linearized, singly integrating, yeast-integrating plasmid with *pADH1-rtTA* and Hygromycin-resistance gene as a selection marker. This yielded two strains, "W303r1" (From 20000A) and "W304r1" (from 20000B). We next replaced the *ade2-1* "ochre mutation" in W303r1 and W304r1 with a functional *ADE2* gene by a homologous recombination of *ADE2* that we obtained by PCR from the S288C reference genome (100-bp homology on both flanking sites of the PCR product). This yielded two strains, "TT2" (from W303r1) and "TT8" (from W304r1). We then inserted a *pTET07-GFP* at the *LEU2* locus in TT2 and TT8 by linearizing a yeast-integration plasmid that contains *pTET07-GFP*. This yielded two strains, "TT7" (from TT2) and "TT9" (from TT8). We then introduced two constitutively expressed selection-marker genes: *URA3* into TT7 to create "TT10" and *TRP1* into TT9 marker to create "TT13". These selection markers allowed us to select the diploid strain ("TT14") that resulted from mating TT10 with TT13 by using a double drop-out medium (lacking -ura and -trp). TT14 is homozygous for *ADE2*, *pADH1-rtTA* and *pTET07-GFP*; thus, all four haploids in a TT14 spore bag has an inducible *GFP*. For the construction of TS3 strain (EU-incorporating diploids), we transformed two haploid strains, "TT10" and "TT13", as stated in (Viggiani & Aparicio, 2006) with a linearized fragment (digested with NheI) of the plasmid "p403-BrdU-Inc" (from Viggiani & Aparicio, 2006 and obtained from Addgene). This plasmid contains *hENT1-ADH1*, *HSV-TKGDP* and a constitutively expressed selection marker, *HIS3*. We linearized and then integrated the plasmid, as a single copy, into the *HIS3* locus in the yeast genome. This resulted in two strains, "TS1" (from TT10) and "TS2" (from TT13), of opposing mating types. We mated these two to obtain a diploid strain "TS3". To construct the strain, "TS8" (which has Rpb3 protein fused to mCherry), we transformed the haploids strains, TS1 and TS2, by integrating mCherry-NatR linker at the *RPB3* locus in the yeast genome. We obtained the mCherry-NatR linker taking a plasmid "HypI" (used in Laman Trip & Youk, 2020 and derived from

"pkt150" (Sheff & Thorn, 2004)) and then amplifying the mCherry-NatR linker through PCR. This resulted in two strains, "TS4" (from TS1) and "TS5" (from TS2), which had opposite mating types. We mated them to obtain the diploid strain, "TS8". For constructing the diploid strain, "TS9" (which has Rpc40 protein fused to mCherry), we transformed the haploids strains "TS1" and "TS2" to obtained two strains, "TS6" (from TS1) and "TS7" (from TS2), which had opposite mating types. We mated them to obtain the diploid strain, "TS9".

## Yeast transformations and mating

We transformed yeasts with the standard, lithium-acetate-based method. In short, log-phase cells were resuspended in 0.1 M of lithium acetate in 1x TE, together with the DNA to be inserted by homologous recombination at a desired location within the yeast genome and herring ssDNA. We then added PEG3350, 10× TE and 1 M of lithium acetate to the yeast culture so that we had final concentrations of 40% PEG 3350, 1× TE and 0.1 M of LiOAc. We then incubated the culture at 42°C for 30 min. We pelleted the resulting cells by centrifuging and then washed them with water. Afterwards, we plated the cells on agar plates with dropout media to select from transformed yeasts. We used a standard method of mating yeasts. Namely, we mated yeasts of opposite sexes by inoculating a 500 μl of YPD medium with two colonies—one from each sex. We then incubated the culture overnight in 30°C on a benchtop shaker (Eppendorf Mixmate) that agitated the culture at 300 rpm. We then spread 100 μl of the culture on double-dropout agar plates which selected for the diploids that resulted from successful mating.

## Spore formation (sporulation)

We used a standard protocol for sporulating yeasts. In short, we first grew diploid yeasts (homozygous diploid wild-type or GFP-inducible strains) to saturation overnight. We then transferred these cells to a "pre-sporulation media" (i.e. YPAc: consists of Yeast Peptone media with a 2% potassium acetate), which we then incubated for 8 h at 30°C. We subsequently transferred the diploid yeasts to a "sporulation medium" (i.e. 2% potassium acetate) and left them to sporulate for 5 days at 20°C, while rotating as a liquid culture in a tube. Afterwards, we transferred the resulting spores to water and stored them at 4°C. Through measurements, we found that we could store these spores for several months without loss of viability (i.e. 2%-glucose still germinates ~100% of the spore bags).

## Microscope sample preparation

We performed all microscope imaging with 96-well glass-bottom plates (5242-20, Zell-Kontakt). Prior to each microscope imaging, we pre-treated the glass-bottom by incubating it for 20 min with 0.1 mM of concanavalin A (ConA, C2010, Sigma-Aldrich). We washed the ConA and then typically added 1 μl of spores in 200 μl of minimal medium per well. The plates were then centrifuged at 150 $g$ for 1 min to sediment and attach all spore bags to the glass-bottom. We then performed the microscope imaging.

## Microscope data acquisition

We used Olympus IX81 inverted, epifluorescence, wide-field microscope. For each time-lapse movie, we collected images once every 10 min for every field of view. The temperature during microscope imaging was maintained by an incubator cage (OKO Lab) that enclosed the microscope. We acquired each image with an EM-CCD Luca R camera (Andor) and IQ3 software (Andor). We used a wide-spectrum lamp (AMH-600-F6S, Andor) for exciting fluorescent proteins.

## Microscope data analysis

We processed the microscope images with ImageJ and MATLAB (Mathworks). To measure the times taken to germinate (and to count spore bags that germinated), we looked for the first haploid spore that formed a bud for each ascus (spore bag). We segmented the spore bags by using the Sobel filtering of the bright-field images to create a mask. We extracted fluorescence values inside the mask. We corrected for the background fluorescence, for each ascus one by one, by subtracting the average background in a 50-pixel area that surrounded each ascus. For GFP fluorescence (measuring GFP inducibility) and mCherry fluorescence (measuring RNAP I-III levels through Rpb3-mCherry or Rpc40-mCherry), we computed the fluorescence per spore by taking the maximal intensity from 36 z-stacks with a z-step size of 0.2 μm per spore bag.

## Hexokinase-based assay to measure glucose concentrations

Ten hours after adding a low concentration of glucose to germinate some of the spores in a 1-ml minimal medium in a 24-well microscopy plate (5242-20, Zell-Kontakt), we took 800 μl of the supernatant to measure the concentration of glucose in it. We determined the glucose concentration by using a hexokinase-based glucose assay kit (Glucose (HK) Assay, G3293, Sigma-Aldrich) that is based on converting glucose through hexokinase and NADP$^+$-dependent glucose-6-phosphate-dehydrogenase.

## RNA-seq on un-germinated spores

We collected un-germinated spores by first collecting 1 ml samples of spores that were incubated in a 0.002%-glucose at 0, 16, 48 and 96 h after the incubation began. To isolate the un-germinated spores, we treated the 1-ml samples with zymolyase (786-036, G-Biosciences). Zymolyase lysed vegetative cells that formed from germinated spores, thus ensuring that we only collected RNA from un-germinated spores for sequencing. We then extracted the RNA from the leftover, un-germinated spores RiboPure Yeast Kit (Ambion, Life Technologies) as described by its protocol. Next, we prepared the cDNA library with the 3' mRNA-seq library preparation kit (Quant-Seq, Lexogen) as described by its protocol. Afterwards, we loaded the cDNA library on an Illumina MiSeq with the MiSeq Reagent Kit c2 (Illumina) as described by its protocol. We analysed the resulting RNA-seq data as previously described (Trapnell *et al*, 2012): We performed the read alignment with TopHat, read assembly with Cufflinks and analyses of differential gene expressions with Cuffdiff. We used the reference genome for *S. cerevisiae* from ensembl. We used the transcriptional modules listed in Appendix Table S1 for grouping the relevant genes into transcriptional modules.

### Fixing and permeabilizing spores

Spores were fixed and made permeable—to let fluorophores into the spores for 5-EU labelling of RNAs and single-molecule RNA FISH—according to the standard protocols for vegetative *S. cerevisiae*. An important modification to these protocols is that to digest the spore wall, we extended the typical incubation in zymolyase to 2 h during which the spores were constantly agitated at 300 rpm in 30°C. We used this fixation and permeabilization treatment for performing single-molecule RNA FISH (Fig 4H) and 5-EU labelling of RNAs (Fig 4C and D).

### FISH probes

We designed single-molecule FISH probes for detecting 18S rRNA (Fig 4H) with the Stellaris® FISH Probe Designer (Biosearch Technologies, Inc., Petaluma, CA): www.biosearchtech.com/stellaris designer. The probes were coupled to CAL Fluor Red 610 (Biosearch Technologies, Inc.).

### Single-molecule RNA FISH

We used the standard protocol for single-molecule RNA FISH in yeast, detailed in "Protocol for S.cerevisiae from Stellaris RNA FISH" (Biosearch Technologies, Inc., Petaluma, CA) and also in Raj *et al* (2008) and Youk *et al* (2010).

### Metabolic (5-Ethynyl Uridine) labelling of freshly synthesized RNA with click reaction

We used click chemistry to bind fluorophores to the 5-Ethynyl Uridine (5-EU) labelled RNAs in spores. To do so, we followed the protocol for mammalian cells, from "Click-iT® Plus Alexa Fluor® picolyl azide toolkits" (Thermo Fisher) but with the following modifications so that the protocol would work in yeast spores: (i) all reactions and washing steps were done in a 1.5-ml Eppendorf tube with a centrifugation speed of 800 *g*; (ii) spore fixation and permeabilization were performed as mentioned above; (iii) total volume for the click reaction was 50 μl; (iv) for the click reaction, fixed spores were at OD ~ 0.02; (v) the click-reaction cocktail was incubated for 1 h and 30 min. All components (Ethynyl-Uridine, Fluorophore-Azide and reagents for the click reaction) were from the "Click-iT® Plus Alexa Fluor® picolyl azide toolkits" (Thermo Fisher). For simultaneously imaging 5-EU bound RNAs with GFP proteins inside spores (Fig 4E), we used Picolyl Azide Alexa 594 from JenaBioscience. Here, the ratio of CuSO4 to the copper protectant was set to 1:20 in order to simultaneously detect the fluorescence from both GFP and Alexa 594.

## Data availability

The authors declare that all data supporting the findings of this study are available within the paper and its supplementary information files. The data that support the findings of this study are available from the corresponding author upon reasonable request. The RNA-seq data in Fig 2G are available on NCBI GEO with accession number GSE159575.

Expanded View for this article is available online.

## Acknowledgements

We thank Sophie van de Gevel for her help with the 5-EU labelling of RNA. We thank Amir Mitchell for insightful discussions and critical comments on our manuscript. We thank the members of the Youk laboratory for fruitful discussions. We thank The Kavli Nanolab Imaging Centre (KNIC) and Jeremie Capoulade for support and help with microscopy. H.Y. was supported by the European Research Council (ERC) Starting Grant (MultiCellSysBio, #677972), Netherlands Organisation for Scientific Research (NWO) Vidi Award (#680-47-544), CIFAR Azrieli Global Scholars Program, and EMBO Young Investigator Award.

## Author contributions

TM and HY conceived the project and designed the experiments. TA and MAB performed the experiments and data analyses for Fig 1. TM, TA and MAB, performed the experiments and data analyses for Fig 2. TM performed all the experiments and data analyses for Figs 3 to 7. HY provided overall guidance. TM and HY wrote the manuscript.

## Conflict of interest

The authors declare that they have no conflict of interest.

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
