## [Review Process File · Molecular Systems Biology]

Dormancy-to-death transition in yeast spores occurs due to gradual loss of gene-expressing ability

Théo Maire, Tim Allertz, Max Betjes, and Hyun Youk

DOI: [10.15252/msb.20199245](https://doi.org/10.15252/msb.20199245)

Corresponding author(s): Hyun Youk (hyun.youk@umassmed.edu)

Review Timeline:

Submission Date:	18th Sep 19
Editorial Decision:	6th Nov 19
Revision Received:	30th Aug 20
Editorial Decision:	15th Oct 20
Revision Received:	20th Oct 20
Accepted:	21st Oct 20

Editor: Maria Polychronidou

Transaction Report:

6th Nov 2019

Thank you again for submitting your work to Molecular Systems Biology. I apologize for the delay in getting back to you, which was due to delays in obtaining the referee reports. We have now heard back from the three referees who agreed to evaluate your study. As you will see below, the reviewers raise a number of concerns, which unfortunately preclude the publication of the study in its current form.

The reviewers mention that i) further analyses and controls would be required to conclusively support the main findings and that ii) the level of insight provided by the study does not seem sufficient for Molecular Systems Biology. However, considering that the reviewers appreciate that the topic and findings seem interesting and therefore likely to be relevant for the field, we would like to offer you a chance to revise the study and address the points raised.

Without repeating all the points/comments listed below, the most fundamental issues that need to be convincingly addressed are the following:

- As reviewer #3 mentions, the addition of further analyses providing some level of mechanistic insight into the link between GFP expression ability and the ability of spores to germinate would significantly enhance the impact of the study.
- Both reviewers #1 and #3 refer to the need to include further analyses and controls to better support the main conclusions.

All other issues raised by the referees would need to be convincingly addressed.

I understand that several of the reviewers' requests may require extensive experimental analyses. In case you would like to discuss an extension of the 90 days deadline for revision please let me know. I would be happy to discuss further any of the points raised by the reviewers (e.g. over the phone), if you think that would be helpful.

On a more editorial level, we would ask you to address the following issues.

REFEREE REPORTS

Reviewer #1:

Summary

This study reports on a novel approach to measuring the dormancy of yeast spores. Yeast cells undergo sporulation in response to nutrient starvation, which results in 4 haploid spores within a spore bag. These spores are thought to be metabolically dormant and physically distinct from vegetative cells. In the submitted manuscript, the authors describe significant cell-to-cell heterogeneity in glucose mediated spore germination, and also how the ability of a spore to germinate decreases with time. The authors hypothesize that the internal state of a spore determines its ability to germinate, and that internal state decays over time. An externally inducible GFP construct is introduced, which is used to show that dormant yeast spores can express GFP to the same level as vegetative cells, and that surprisingly, an individual spore's GFP intensity corresponds to its germination probability. The proposal of a "dormancy spectrum" to measure the transition from dormancy to death was conceptually novel.

General Remarks

The manuscript is interesting, but has a problem with having much grander statements in the introduction and discussion than what is presented in the results. A major concern is that what is observed in this study may only apply to yeast spores, which may not be a good model for microbial dormancy, and also the results may be specific to the experimental setup. A major revision is recommended.

Major Points

This study posits that yeast spores are dormant cell structures 'ceasing life', while finding that spores can express GFP from an externally inducible gene circuit. The authors cite several papers (11-12,16-21 in the manuscript) to support this claim of yeast spore dormancy. However, many of the cited studies are works on bacterial cells or spores, which have major differences from yeast spores in structure, composition, and function. Indeed, in one of the papers cited in the paper (12), the main finding is that they find ongoing transcription and translation in nongerminating yeast spores incubated in water at 30 {degree sign}C. In accordance with this, the current manuscript describes that yeast spores incubated in water at 30 {degree sign}C slowly loses its ability to germinate, with complete loss occurring as early as 80 days. This calls into question how the claims of yeast spore dormancy can be compared to bacterial spores, which can be dormant for millions of years. As yeast spores are thought to evolutionarily arise from the need to withstand ingestion from fruit flies, yeast sporulation should be thought of a dispersion method rather than a survival method. At best, this indicates that yeast spores are a fragile form of dormancy.

The finding that ungerminated yeast spores are able to express GFP from an inducible gene circuit, and that spores gradually lose their ability to germinate is interesting. However, this raises several questions that are inadequately addressed in the manuscript.

- (1) If doxycycline is able to transverse the spore wall, this alone indicates that the yeast spore is far more permeable compared to bacterial spores.
- (2) If GFP is able to be expressed, this shows that the cytoplasm of the yeast spore is less rigid than bacterial spores, which are in an immobile and dehydrated state to prevent enzymatic activities. The claims throughout the manuscript that yeast spores are in a state of ceased life comparable to bacterial spores is questionable and need substantive rephrasing.
- (3) What exactly the GFP intensity is measuring is unclear. The manuscript does not address this, apart from alluding to abundances of stored ribosomes or RNA polymerase without experimental evidence. This needs much more detail as GFP fluorescence as a proxy for germination potential is the central observation in this study.

(4) As previous studies indicate yeast spores continue metabolic activities while incubated in 30C water, the fact that dormant spores slowly cease to germinate seems to be a result of the experimental setup, not a general characteristic of dormancy as claimed in the introduction (Lines 69-73). This limitation was not clear from the manuscript. In addition, the statement on yeast spores retaining germination ability for "several months" in 4C water (Lines 649-651) needs more detail. Perhaps repeating the experiment in a much lower temperature may yield insight.

(5) Line 430: "Dormant spores would gradually lose their ability to express genes without nutrients due to, for example, their intracellular components naturally (thermally) degrading over time" is a statement that is unsupported by evidence in the manuscript and too broad. Is this a general characteristic of microbial spores? Is there evidence that internal components are thermally degraded?

(6) The study asserts that spores unable to germinate in saturating (2%) glucose are dead. This is a strong statement, and needs further clarification. Are there any known germinants other than glucose? What happens when you add another pulse of 2% glucose, much like the experiments in Fig 2? Are they unable to form colonies on rich media agar plates?

Minor Points

1. A more thorough description of the yeast spore would be desirable. The current introduction makes broad associations with results from bacterial spores, which makes understanding the results in this paper more difficult.
2. Figure 1b needs a scale bar and time stamps.
3. Figure 2: The results on primed dormancy do not have a strong association with the rest of the manuscript and seem to be a sidebar. Reorganizing it to put more emphasis on the motivation of the experiment (showing that ungerminated spores are not necessarily dead) would streamline the paper.
4. Figure 2b Y-axis: is this a percentage of the total spores measured? The numbers do not add up with the plot in 2c upper row.
5. Fig 3c: the two plots should have matching Y-axis ranges.
6. The paragraph starting from line 344 is perhaps too dramatic and can be omitted.
7. Line 362: "rather than a qualitative relationship such as 'spores that can express GFP can germinate whereas those that cannot express GFP do not germinate'." is unnecessary.
8. Line 568: comparison to plant seeds is a stretch because seeds pack nutrients that warrant initial growth.
9. There are many instances of grammatical errors throughout the manuscript, which do not necessarily take away from the substance of the paper, but will benefit from a professional editing service.

Reviewer #2:

Summary

This study focuses on the dividing line between dormancy and death. Using yeast spores as a model for dormancy, the authors explore how dormant lifeforms can either be revived or be unrevivable. Combining systems and synthetic biology approaches, they identify how a spore's gene expression level is related to the level of glucose required to germinate later and how likely germination is instead of death.

General Remarks

This interesting study focuses on defining death vs. dormancy and what factors differentiate dormant spores that can and cannot be revived. This is an exciting and relatively unexplored question and the authors take a very creative approach to tackling this conundrum. The study's key finding that some global control mechanism of transcription and translation is linked to revival potential for spores is very exciting and will be of broad interest to the microbial research community. However, one concern is that the study design may potentially conflate the ability to sense the environment while dormant (sensing doxycycline and glucose) with the ability to express GFP when dormant and revival ability. While these factors are highly linked, because if you cannot sense the environment you cannot receive the signal to express GFP or be revived by glucose, some commentary by the authors on this point would strengthen the manuscript. For example, the priming phenomenon discussed might be that sensing some glucose induces spores to upregulate their ability to sense environmental changes in general. This could assist with their eventual revival and shorten the time required to sense and respond to these changes, leading to a shorter revival time.

Minor points

- Abstract (lines 33-34) and other locations: "How organisms with ceased lives remain alive and what sets their lifespans are fundamental questions." The idea of ceased life being the same as dormancy appears here and in other locations throughout the manuscript. The idea of ceased life implying dormancy and not death could be confusing to readers who think of ceased life as death. It may be helpful to explicitly define dormancy, ceased life, and death in the manuscript as these terms are used frequently and have specific meanings that are explored and challenged throughout the text.
- Lines 197-199: "These results establish that spores that do not germinate after encountering ample glucose are not necessarily dead." The timing of the ample glucose exposure should be clarified in this sentence as this section is about priming germination with an initial encounter with glucose and then a second, subsequent encounter.
- The section "Synthetic circuit shows that activating GFP expression in dormant yeast-spores does not alter percentages of spore bags that germinate" discusses an important control for the synthetic biology approach to understanding what controls the ability of dormant spores to germinate, but may be best moved to the supplement.
- It could be exciting in follow-up work to explore the implications of priming as a phenomenon specifically to see if arbitrary gene expression ability changes after priming. This is implied by the upregulation in gene expression across most genes in Figure 2g, but isn't explicitly revisited after the synthetic GFP expression circuit is introduced.
- General comment: The way Glucose % is displayed in figures should be consistent throughout the manuscript. For example, in Figures 1 and 2, Glucose % is displayed in scientific notation, but in Figure 3, decimal notation is used.

Reviewer #3:

Spores are typically thought to be in a state of metabolic inactivity, where a glass-like cytoplasm preserves cellular integrity. This results from both direct evidence in some species and the fact that spores can remain viable for many years; thus, it seemed unclear how any spores that had

metabolic activity could remain viable. Contrary to this, a small amount of work has suggested that, at least in some species, spores have metabolically activity. The authors here convincingly show a surprising new aspect of the spore physiology: yeast spores are able to express genes to a significant level, as shown by the expression of GFP from a synthetic construct. Even more surprising, the GFP expression level correlates with the probability of a spore to germinate. With this system, the authors characterize the relationship between germination requirements and GFP expression. They find that yeast spores require a certain concentration of glucose to germinate. When expressing GFP from an inducible promoter, higher GFP expression levels lower the concentration of glucose required to sporulate. In addition, over time they find that spores express less GFP, and cells require higher glucose concentrations to germinate. From this finding, the authors draw the conclusion that yeast spores lose the ability to germinate, because they eventually require a concentration of glucose higher than what they can be exposed to (>2%).

The interesting phenomenon observed by the authors, that GFP expression correlates with the probability to germinate, by itself, I believe, is insufficient for publication in *Molecular Systems Biology*. Several of the figures in the manuscript, such as the double pulses of Fig. 2 or the model of Fig. 4 do not help in strengthening the key point of the paper. The minimal amount of glucose needed to germinate (Fig. 1&2) and the ability to express GFP (Fig. 3&4) are not causally linked in this manuscript, other than by a model that is mathematically describing the correlation. Taken together, I cannot currently advice publication of the manuscript.

I do believe, however, that by probing what the GFP expression ability means, why it is decreasing and why this is important for maintaining viability, the authors could eventually bring this manuscript on a level appropriate for *Molecular Systems Biology*. The basic phenomenon reported in this work is interesting and important for the community, and I encourage the authors to probe deeper for the molecular mechanism. I provide some feedback below. I do not expect the authors to perform all of these experiments, and I do not know, if all experiments are performed, whether this manuscript will be sufficient for publication in *Molecular Systems Biology* as it is hard to predict the results. If the authors decide to resubmit this paper, I would be happy to re-review this manuscript.

Major points

- 1) The authors' argument is based on two correlations. First, GFP expression levels correlated with the minimum glucose concentration required for sporulation. Second, GFP expression levels anticorrelated with time. Combining these two correlations will result in a new correlation: the GFP expression ability at one point in time will correlate with its life time. While plausible, the authors would need to show 1) that spores with a higher GFP expression ability will actually live longer (I suspect that the authors are right, but they need to provide experimental evidence for this claim); and 2) the authors would need to show experimental evidence that glucose is limiting, as the authors claim that spores lose the ability to germinate because they cannot be exposed to high enough glucose concentrations.
- 2) Why does higher GFP expression lead to higher germination rates? This is the central point where the manuscript must be improved to be publishable in a journal like *Molecular Systems Biology*. Understanding what GFP expression ability is a proxy for can give the authors valuable insights into why spores are dying. Some progress needs to be made. One option is that GFP expression ability correlates with the internal energy state of the spore. One way to test this, could be to add ionophores like DNP or CCCP to inhibit ATP production and to see if their presence during dormancy will shorten the life span of the spores. Another approach could be to block transcription/translation and test if protein synthesis during dormancy is important for survival. Adding nutrients, glucose or other carbon substrates, could replenish the energy pool (if limited), and increase GFP expression. Will it also increase life span? In case spores are energy-limited,

expressing luciferase prior to sporulation could allow the authors to get a direct measurement of the ATP content in the spores.

3) Directly measuring life span as a function of GFP expression levels. The authors have good evidence that the GFP expression ability at given point in time correlates with their ability to germinate. The authors are missing an experiment that shows that higher GFP expression predicts also correlates with germination efficiency at a later time. The authors could induce GFP expression for a limited time, e.g. for 1 day early in sporulation, and then measure the germination probability several days later.

4) Model of the increase of minimal glucose concentration. According to the calculation of Fig. 4f, spores with low GFP expression ability increase their minimal glucose concentration at a rate of less than $1.5 \cdot 10^{-3}\%/d$. At this maximal rate, after 40 days, I expect the minimal glucose concentration to be $40d \times 1.5 \cdot 10^{-3}\%/d = 0.06\%$, well short of the 2% maximal glucose concentration of the experiments. This is obviously not what the authors' model produced. The authors' model seems instead to be built around the decrease of GFP expression ability from a finite number down to zero. Once the cells hit zero, see Fig. S24d, spores are pronounced dead. The model is missing an obvious test: does the estimated life span of Fig. 4g correctly predict the survival curve of Fig. 4a? Apart from this, the model makes sense, but it took me an awful long time to understand it. If I interpret it correctly, with the model the authors can discuss the kinetics of death. An aspect that is hidden in the experimental data of Fig. 4c, which only considers viable spores. It was unclear to me, why the authors dwell on the minimal amount of glucose to wake-up, Fig. 4f. The loss of GFP expression ability seems to be clearly correlated with survival and a much more physiological meaningful parameter. The minimal amount of glucose to wake-up diverges at the lowest GFP expression ability in Fig. 4e, making it hard to measure and hard to interpret. It is furthermore not clear whether the model is truly describing the minimal amount of glucose to wake-up. Rephrasing the model in terms of GFP expression ability could help it describing a more meaningful aspect of the spores' life.

Minor points

1) Priming with low glucose concentration. Fig. 2 seems misplaced in the story of the paper. Why prime the spores with glucose, but draw no conclusion from the result? I would suspect that the GFP expression ability changes after a low glucose pulse. If the GFP expression ability changes, the authors have a very beautiful system at hand to test if the lifespan of the spores, changes, also.

2) Are all yeast sporulated. I assume that the authors have a way to ensure that all yeast are indeed sporulated, or at least that only sporulated yeast are included in the data analysis. I would like to see evidence of this.

3) Glucose concentration range. In experiments such as Fig. 4c, the authors show a very narrow concentration range, while Fig. 4e shows a different range. Changing glucose concentration ranges makes it hard to interpret data (e.g. from 4c to 4e). Why the authors focus on the range of 0.001% to 0.01% to make the point that the minimal glucose concentration to germinate is increasing above 2% is confusing. What about the intermediate regime?

4) 99% germination metric. From germination landscapes like Fig. 4c, the authors extract the minimum concentration of glucose to wake up 99% of spores. It is unclear how the authors got this metric. A fit of a mathematical model? Or the lowest experimental concentration where more than 99% sporulated? Either way, this seems like a dangerous metric, that could depend on the detailed choice of the model, or the experimental noise. The authors should show that their results are robust to other choices of metrics.

5) Synthetic promoter. Spore bags induced with doxycycline plateau at a certain expression level. It is a little surprising given that there is no cell growth that level don't continue to increase over time. Doxycycline is unstable and potential is functional depleted in the experiments by the end of one day. It would be good for the authors to swap media with fresh doxycycline once per day to ensure

their results are not influenced by doxycycline breakdown.

For the past nine months - before and during the COVID-19 restrictions in the Netherlands - the first author (Théo Maire) steadily worked to mechanistically explain the phenomenological findings from our original manuscript. On pgs. 1-3 of this letter, we summarize the additional findings before giving point-by-point responses to the reviewers.

Summary of the original manuscript

Our original manuscript showed that a synthetic circuit can induce as much GFP expression in dormant yeast spores as in vegetative yeasts, despite the spores being in water or PBS without any nutrients. We then showed that if a dormant yeast spore can make more GFP without nutrients (i.e., if the spore has higher "GFP inducibility"), then it has a higher probability of germinating (i.e., higher "germination ability") for any glucose concentration and requires less glucose to guarantee that it germinates (i.e., 100% chance). Finally, we used a mathematical analysis to argue that a dormant yeast spore with a higher germination ability lives longer. Hence, the main phenomenon was that *GFP* expression predicts the spore's lifespan and germination ability (i.e., probability of germinating for each [glucose] and the minimum [glucose] required to guarantee a germination). The question then is why GFP, a protein without any crucial function in yeast, can so precisely predict both the lifespan and germination ability. The manuscript ended by saying that GFP represents the spore's gene-expressing ability during dormancy but did not explore it further.

Summary of additional conclusions from the revision and connections among them

Our original conclusions remain true. We obtained additional results, including molecular explanations of the GFP inducibility, that expand the original conclusions. These results are:

1. Inhibiting ATP synthesis with a drug (antimycin A) for 24 hours does not affect the spores' germination ability (i.e., nearly 100% of the spores can still germinate after removing the drug). --- See Fig. 4B.
2. Inhibiting translation with a drug (cycloheximide) for 24 hours does not affect the spores' germination ability. --- See Fig. 4B
3. Inhibiting transcription with a drug (thiolutin) for 24 hours kills nearly all spores. --- See Fig. 4B
4. Result #3 is likely not because making of all transcripts is important for spores to survive. This is supported by the fact that spores' total transcription rates poorly correlate with their GFP inducibility (and thus poorly correlates with their germination ability). --- See Fig. 4E.
5. Supporting the importance of transcription, amount of RNA polymerase II (proxied by fluorescence of mCherry protein fused to Rpb3 - a subunit of RNAP II) highly and positively correlates with the GFP inducibility and thus with germination ability. --- See Fig. 4F.
6. Like RNAP II, amount of RNAP I & III combined (proxied by fluorescence of mCherry protein fused to Rpc40 - a subunit of both RNAPs I & III) highly and positively correlates with GFP inducibility and germination ability. --- See Figs. 4G & 4I.
7. Unlike RNAPs, amount of ribosomes (proxied by the amount of 18S rRNA - a subunit of the ribosome) is virtually uncorrelated with GFP inducibility. --- See Fig. 4H

8. Combining #5 & #6 with pre-revision results (Fig. 3) leads to a triangular relationship in which a pair from any of the following three metrics are highly and positively correlated: Germination ability for each glucose concentration, amount of RNA polymerases, and GFP inducibility. Hence RNAPs act as molecular/mechanistic windows into the correlation between GFP inducibility and germination ability. Of the three RNAPs, we focused on RNAP II since it transcribes all coding genes and makes some non-coding RNAs ---- See Fig. 4J
9. By incubating spores in water at 30 °C and then measuring their RNAP II levels over ~2 months, we discovered how dormant spores age: RNAP II level and GFP inducibility both decrease over days. Both eventually reach nearly zero values of dead spores. -- See Figs. 5B-C
10. Moments before dying, aged spores that are alive (i.e., dormant) have nearly undetectable RNAP II levels like dead spores but they still can, after receiving a saturating concentration (2%) of glucose, make more RNAP IIs whereas dead spores cannot. --- See Fig. 5F
11. Combining #8 - #10: dormant spores age by losing their gene-expressing ability and die once they have sufficiently lost the ability, which even a 2%-glucose cannot restore - these spores cannot express, for example, the genes necessary to make more RNAP II.
12. To further explore #11 in light of #3: inhibiting transcription with a drug (thiolutin) - for a day or less - causes dormant spores to permanently lose their GFP inducibility, meaning that removal of thiolutin does not restore their GFP inducibility to the same level that it initially was. Crucially, a day-long inhibition completely eliminates the GFP inducibility and kills all spores. -- See Fig. 6B.
13. Despite #12, all spores still have plenty of RNAP II - more than half the amount of RNAP II that they started with - after 24 hours of transcription inhibition by thiolutin. Yet, after receiving a 2%-glucose, all these spores do not make more RNAP II and are dead. -- See Fig. 6E.
14. Combining #12 & #13: If spores cannot transcribe for a day, they permanently lose key molecules that are required for gene expression, other than RNAP II, on a timescale of at most 12-to-24 hours, causing them to die. We argue that these are likely key transcripts that are required for spores to maintain their gene-expressing ability during dormancy.
15. Motivated by #14: As an example, we focused on a macro-molecule that is central to gene expression - RNAP II - and showed that, during dormancy, spores produce RNAP II extremely slowly (characteristic production time of ~25 days) compared to the rate at which RNAP II degrades (half-life of ~8.4 days in spores). Hence, there is a net loss of RNAP II over time during dormancy, inevitably causing RNAP II level to reach virtually undetectable levels (explaining #9 & #10) at the last moments of dormancy before the spore dies. --- See Figs. 7B-E
16. Combining all the findings culminates in a new conceptual model of how dormant yeast spores age and eventually die after enough time without nutrients: key molecules that are required for gene expression, including RNA polymerases, are gradually lost during dormancy because spores cannot produce them faster than they degrade - we measured firm numbers (tens of days) for these extremely slow processes. These molecules eventually reach below some threshold amounts after which the spore has permanently lost the ability to express genes, meaning that even ample glucose cannot restore the

gene-expressing ability. Since, after receiving glucose, a spore must express genes in order to build a new cell that will bud off of the spore (i.e., germinate), the permanent loss of gene-expressing ability at the final moments of dormancy marks the spore's death. To speculate, we believe that some of these key molecules required for maintaining gene-expressing ability are transcripts - perhaps non-coding RNAs - given our results on transcription inhibition and the fact that proteins like the subunits attached to RNAP II likely degrade on the order of tens of days, not hours (as hinted by thiolutin experiments). We end with a discussion of this possibility, which invites future investigations. --- See Fig. 7F.

List of new figures, experiments, methods, and strains

- Added 4 new main figures (Figs. 4-7) to replace the older Fig. 4 (mathematical analysis of ageing). The new figures reveal the mechanistic picture of ageing and death in dormancy. They negate the more phenomenological, mathematical analysis of the old Fig. 4.
- Added 8 new Appendix Figures (previously called "supplementary figures" but now renamed to meet MSB's style). We now have "Appendix Figs S1-S33".
- Added experiments with drugs that inhibit ATP synthesis, translation, and transcription.
- Added experiments with new strains. Of these, the two most important ones are:
 - Strain with RNAP II made fluorescently visible due to mCherry protein fused to Rpb3, a subunit of RNAP II. It also has the GFP-inducing synthetic circuit. We verified that fluorescent RNAP IIs localize inside the nucleus. Thus, by measuring the mCherry fluorescence in each spore with time-lapse microscopy, we inferred the total abundance of RNAP II in each spore over time.
 - Strain with both RNAPs I and III made fluorescently visible due to mCherry protein fused to Rpc40, a subunit of both RNAPs I and III. It also has the GFP-inducing synthetic circuit. We verified that the fluorescent RNAPs localize inside the nucleus. Thus, by measuring the mCherry fluorescence in each spore with time-lapse microscopy, we inferred the total abundance of RNAPs I & III combined in each spore over time.
- Added new ageing experiments in which drugs (thiolutin or cycloheximide) are present.
- Added experiments in which we used single-molecule RNA FISH to measure ribosomal RNA (18S rRNA: subunit of ribosomes), which is a proxy for the amounts of ribosomes in dormant spores, with a single-cell resolution.
- New method for yeast: we modified an existing method for mammalian cells - metabolic labelling of newly synthesized RNAs with 5-Ethynyl Uridine (5-EU) - and adapted it to yeast. With this method, we quantified freshly made RNAs in dormant yeast spores with single-cell resolution. This is, to our knowledge, one of the first demonstrations of active transcription in dormant yeast spores at the single-cell level.

In the next pages, we provide point-by-point responses to the reviewers' comments.

Response to Reviewer 1:

We thank Reviewer 1 for his/her positive comments regarding the conceptual novelty and interest of our work. We also thank him/her for prompting us to more explicitly state that our findings on yeast spores do not necessarily apply to other spores, particularly bacterial spores. It was not our intention to tell the readers that our results on yeast spores automatically apply to bacterial spores. Our manuscript did not state this. But we can now see that parts of the original manuscript may be misconstrued. To avoid such misunderstandings, we have now revised the introduction and discussion sections and reworded the results section. The introduction section now focuses on yeast right away, though it still mentions some bacterial work to give context. The discussion section now has a paragraph dedicated to saying that our results don't necessary apply to prokaryotic spores. This paragraph contains the sentence, "*Our discoveries on yeast spores may not apply to bacterial spores such as B. subtilis spores and determining the similarities and differences between dormant state of yeast spores with dormant states of bacterial spores is challenging because dormancy of both remain incompletely understood.*"

Below, Reviewer 1's remarks are in *italics* and are followed by our responses.

"Summary

This study reports on a novel approach to measuring the dormancy of yeast spores. Yeast cells undergo sporulation in response to nutrient starvation, which results in 4 haploid spores within a spore bag. These spores are thought to be metabolically dormant and physically distinct from vegetative cells. In the submitted manuscript, the authors describe significant cell-to-cell heterogeneity in glucose mediated spore germination, and also how the ability of a spore to germinate decreases with time. The authors hypothesize that the internal state of a spore determines its ability to germinate, and that internal state decays over time. An externally inducible GFP construct is introduced, which is used to show that dormant yeast spores can express GFP to the same level as vegetative cells, and that surprisingly, an individual spore's GFP intensity corresponds to its germination probability. The proposal of a "dormancy spectrum" to measure the transition from dormancy to death was conceptually novel."

We thank Reviewer 1 for this positive overall assessment.

"General Remarks

The manuscript is interesting, but has a problem with having much grander statements in the introduction and discussion than what is presented in the results. A major concern is that what is observed in this study may only apply to yeast spores, which may not be a good model for microbial dormancy, and also the results may be specific to the experimental setup. A major revision is recommended."

The discussion section now explicitly states that our results are only for yeast spores and that one needs further studies to determine whether these results also apply to other eukaryotic or bacterial spores. We never intended to and did not claim that our results automatically apply to all microbial spores (particularly to bacterial spores).

The introduction section still describes bacterial spores because we believe that it is important to state what is known about other microbial spores, to place our work in the context of previous studies on eukaryotic and prokaryotic spores. The cited studies tend to be on bacterial spores, in particular *B. subtilis* spores, rather than yeast or other eukaryotic spores. This is because there are a lot more systems-level studies of bacterial spores than eukaryotic spores. Compared to the wealth of literature on bacterial (*B. subtilis*) spores, relatively little is published on the dormant state of yeast spores itself. A lot is known about how starved yeasts sporulate and how the yeast spores germinate *after* receiving external glucose. But little is known about what, if any, occurs within the dormant yeast spore itself. Hence, we cannot state whether many well-established aspects of bacterial spores apply to dormant yeast spores or not - we simply do not know enough about dormant yeast spores.

Regarding the reviewer's statement, "...yeast spores, which may not be a good model for microbial dormancy", we respectfully disagree and the majority of yeast biologists, we believe, would also disagree. Among single-celled eukaryotes, *S. cerevisiae* has been a model organism for studying eukaryotic dormancy among microbes. In fact, one of the first microarray-based studies examined the genetic program that controls the yeast's sporulation (cited in our paper: Chu et al. 1998). Studies have also examined germination - the process occurring *after* receiving ample glucose (only 2%-glucose used prior to our work) - and have revealed insights on genetic control of spore's exit from dormancy (cited in our paper: Joseph-Strauss et al. 2007). We believe that studying both bacterial spores and eukaryotic spores - yeast spore is a prime example of eukaryotic spores - is important for broadening our understanding of microbial dormancy by, for example, revealing how eukaryotic spores are different from bacterial spores.

Regarding our "experimental setup", we studied the most commonly studied strain of yeast ("wild-type" strain) and its relatives (e.g., strain with a synthetic gene circuit and, in the revision, strains with fluorescent RNA polymerases). We studied their dormancy at 30 °C. This is the usual temperature at which one studies *S. cerevisiae*. We aged the spores in water or PBS without any nutrients at 30 °C. These are the most general conditions that one can think of. Indeed, our results may not apply to dormant yeast spores at other temperatures. In that sense, our results are specific to our experimental set up - one that many yeast biologists use for studying yeast. But aside from the incubation at 30 °C, there is nothing especially restrictive or specialized about our experimental setup. Our results also do not depend on the yeast having a synthetic circuit (as we also study wild-type strain and show that it exhibits the same primary behaviors as the engineered strains).

"Major Points

This study posits that yeast spores are dormant cell structures 'ceasing life', while finding that

spores can express GFP from an externally inducible gene circuit. The authors cite several papers (11-12,16-21 in the manuscript) to support this claim of yeast spore dormancy. However, many of the cited studies are works on bacterial cells or spores, which have major differences from yeast spores in structure, composition, and function. Indeed, in one of the papers cited in the paper (12), the main finding is that they find ongoing transcription and translation in nongerminating yeast spores incubated in water at 30 {degree sign}C. In accordance with this, the current manuscript describes that yeast spores incubated in water at 30 {degree sign}C slowly loses its ability to germinate, with complete loss occurring as early as 80 days. This calls into question how the claims of yeast spore dormancy can be compared to bacterial spores, which can be dormant for millions of years. As yeast spores are thought to evolutionarily arise from the need to withstand ingestion from fruit flies, yeast sporulation should be thought of a dispersion method rather than a survival method. At best, this indicates that yeast spores are a fragile form of dormancy."

Although there is no debate in the yeast community on whether or not *S. cerevisiae* spores are dormant, the reviewer touches on the important question of what "dormancy" actually means. Currently, there is no universally accepted, quantitative definition of dormancy. Indeed, even for the "canonical" bacterial endospores such as *B. subtilis* spores, an open question is how "silent" the dormant bacterial spore is. For example, the introduction section of the paper on *B. subtilis* spores that cited (Segev et al. 2012) states: "*It is still disputed whether the spore is entirely dormant or whether it maintains some metabolic processes. In fact, it has been shown that bacterial spores do contain certain enzymatic activities; full maturation of coat proteins was found to be dependent on enzymatic reactions taking place subsequently to release of the spore from the mother cell.*"

Our central idea is that, as the reviewer and the quote above alludes, dormancy is *not* a binary phenotype (i.e., dormant or vegetative), at least for yeast spores. Quantifying the degree of dormancy at a systems-level is an underexplored idea. As our manuscript states, our work's primary goal is establishing a systems-level metric for quantifying the degree of dormancy in yeast spores. In our original manuscript, we uncovered the "dormancy spectrum" as a quantitative measure of yeast spore's dormancy. Clearly, yeast spores are not the same as the constantly active, replicating, vegetative yeasts. On the other hand, as the reviewer points out and as our work shows in detail, yeast spores are capable of expressing genes without nutrients. To what degree one can silence the internal activities inside a yeast spore before it can no longer germinate (i.e., dead) is a topic that our work quantitatively addresses. As the discussion section explains, our work quantitatively showed how yeast spores gradually lose the "amount of dormancy" - a quantity that can be proxied by the systems-level metrics that we uncovered in our work: germination ability, amounts of molecules such as RNAP II that are required for gene expression, and the ability to express genes during dormancy when forced to (e.g., GFP inducibility).

The reviewer mentions several points above. To address them in detail, we break them down one-by-one and address each of them below.

- 1) *“However, many of the cited studies are works on bacterial cells or spores, which have major differences from yeast spores in structure, composition, and function.”*

We still cite the bacterial work for context. Discussion section now explicitly mentions that one cannot blindly translate our results to bacterial spores. As we mentioned above, there are a lot more systems-level studies of bacterial spores than eukaryotic spores. Compared to the wealth of literature on bacterial (*B. subtilis*) spores, there are relatively few papers published on the dormant state of yeast spores (i.e., not sporulation or germination). Consequently, many questions regarding dormant yeast spores were unanswered prior to our work. Many of them still remain unanswered. This means that comparing yeast spores with bacterial spores is challenging. Given this state, we believe that our study is a valuable contribution.

-
- (2) *This calls into question how the claims of yeast spore dormancy can be compared to bacterial spores, which can be dormant for millions of years.*

Bacterial spores kept in a salt crystal or ambers are indeed thought to be able to remain dormant (retain replicability) for long periods of time. But we are not aware of papers that experimentally prove that they can be dormant for millions of years.

Our work on yeast spores stands on its own whether or not our results also apply to bacterial spores. But both yeast and bacteria form spores, and both face the same type of problem: how to retain the ability to self-replicate and re-enter an active (vegetative) form of life once nutrients reappear. While we agree that bacterial spores and yeast spores may use different molecular mechanisms to remain viable without nutrients, many properties of dormant yeast spores remain unknown so that we cannot definitively compare dormant yeast spores with bacterial spores. We believe that we should clearly state an honest assessment of the situation - that one does not know enough about dormant yeast spores to clearly distinguish them from bacterial spores. The discussion section now includes this assessment.

In our experiments, the relatively “short” lifespan (up to ~2 months) comes from the fact in our “ageing experiments”, we kept the spores in warm water (at 30 °C) without nutrients for up to 2 months (e.g., in Fig. 5). But one can desiccate yeast spores - and vegetative yeasts - so that they become powders, which are thought to survive for years (though such colloquial statements as “surviving for years” are often more anecdotal than conclusions based on time-lapse measurements over tens of years). Moreover, it is well known - we also see for ourselves despite not reporting it in our manuscript - that yeast spores can remain alive in the fridge (at ~ 4 °C) for at least six months (longer than at 30 °C). Hence, in different incubation conditions, yeast spores can live longer than we reported.

(3) As yeast spores are thought to evolutionarily arise from the need to withstand ingestion from fruit flies, yeast sporulation should be thought of a dispersion method rather than a survival method.

The reviewer raises an interesting hypothesis for why yeast sporulation arose through evolution and the main purpose of sporulating for *S. cerevisiae*. But the evolutionary need and purpose of forming a spore – whether it is for survival or dispersion - are not our focus despite them being interesting topics on their own. We do not claim anything about the purpose and evolutionary origins of yeast spores. We simply focus on the mechanistic (intracellular) aspects of yeast spore dormancy.

(4) At best, this indicates that yeast spores are a fragile form of dormancy.

Papers on yeast spores and the general consensus among yeast biologists is that yeast (and fungal) spores are dormant. But as we stated above, the reviewer touches on the important question of what “dormancy” actually means and there is currently no universally accepted, quantitative definition of dormancy. Precisely because of this, we sought to address how close to being active (replicative) and being dead the yeast spores are. We uncovered at least three different continuous metrics of dormancy which can be considered as “amount of dormancy” as we explain in our manuscript. Consequently, advancing the idea that - as the reviewer alludes - dormancy is not a simple binary phenotype (dormant or not dormant). We also agree with the reviewer’s general sentiment, expressed in his/her series of comments above, that the molecular mechanisms that control the germination and formation of yeast spores are different from those that control the germination and formation of bacterial spores. In the revised discussion section, we now explicitly say that our findings do not necessarily extend to bacterial spores because of this.

"The finding that ungerminated yeast spores are able to express GFP from an inducible gene circuit, and that spores gradually lose their ability to germinate is interesting. However, this raises several questions that are inadequately addressed in the manuscript.

(1) If doxycycline is able to transverse the spore wall, this alone indicates that the yeast spore is far more permeable compared to bacterial spores."

Bacterial spores are also permeable to small molecules (e.g., see Black & Gerhardt, "Permeability of bacterial spores. I. Characterization of glucose uptake". *J Bacteriol.* 1961). Doxycycline, a small molecule, can indeed diffuse through the spore wall. But it does not mean that yeast spores are therefore not "dormant". First, as we stated above, our central idea is that dormancy for yeast spores is a spectrum rather than a binary phenotype. Secondly, doxycycline diffusing through the spore wall does not mean that all small molecules can freely diffuse through the yeast spore wall. For example, both vegetative yeasts and yeast spores can only uptake glucose by using a passive transporter (hexose transporters called Hxts). Knocking out enough *HXT* genes is lethal for *S.*

cerevisiae. Glucose cannot readily diffuse through even the vegetative yeast's cell wall, which is thinner than yeast spore wall (Neiman, 2011), to the extent that the vegetative yeasts cannot bring in sufficient amounts of glucose just by free diffusion. The cell needs at least one of the Hxts. Given this, it is unlikely that extracellular glucose can freely diffuse through the thick spore walls to the extent that spores would be metabolizing glucose without any Hxts. But in the end and most importantly, whether or not yeast spores are more permeable to small molecules than bacterial spores is not the issue as that alone does not determine whether a spore is dormant or not dormant, as we explained above.

"(2) If GFP is able to be expressed, this shows that the cytoplasm of the yeast spore is less rigid than bacterial spores, which are in an immobile and dehydrated state to prevent enzymatic activities. The claims throughout the manuscript that yeast spores are in a state of ceased life comparable to bacterial spores is questionable and need substantive rephrasing."

As we originally stated in our introduction (and still there), a recent paper that we cite shows that dormant yeast spores also exhibit a glassy (solid-like) cytoplasm that, as in bacterial spores, drastically reduces diffusion (Munder et al, "A pH-driven transition of the cytoplasm from a fluid- to a solid-like state promotes entry into dormancy" *Elife*. 2016). In it, the authors state, "*Here, we study dormancy in different eukaryotic organisms and find it to be associated with a significant decrease in the mobility of organelles and foreign tracer particles. We show that this reduced mobility is caused by an influx of protons and a marked acidification of the cytoplasm, which leads to widespread macromolecular assembly of proteins and triggers a transition of the cytoplasm to a solid-like state with increased mechanical stability. We further demonstrate that this transition is required for cellular survival under conditions of starvation.*"

Conversely, another paper that we cited (Segev et al, *Cell* 2012) shows that *B. subtilis* spores actually exhibit traces of metabolic/enzymatic activity despite the physical properties of their cytoplasm that the reviewer mentions.

In light of the findings stated above, an interesting question is: how can dormant yeast spores still express *GFP* at nearly the same high levels as vegetative yeasts - a surprising finding in our work - despite the spore's cytoplasm having physical characteristics, like bacterial spores, that drastically limit diffusion and enzymatic activities? This is in fact the question that we posed in the discussion section in our original manuscript (and still do). This question - how the central dogma of molecular biology works in dormant yeast spores - now makes sense to ask because of our discovery that fully inducing gene expression at the level of vegetative yeasts is possible (and thus, the question is why is a dormant yeast spore barely expressing any genes as shown in one of the papers that we cite (Berengues et al. 2002)? Moreover, our additional experiments from the revision show that dormant yeast spores have highly stable RNA polymerases (RNAP I-III) - their half-life is ~10 days - and that the dormant spores are making new RNAPs, albeit extremely slowly. Addressing this question is beyond the scope of our current manuscript, which deals with

quantifying determinants of germination and death during dormancy. The discussion section describes potential future studies for addressing this question.

“(3) What exactly the GFP intensity is measuring is unclear. The manuscript does not address this, apart from alluding to abundances of stored ribosomes or RNA polymerase without experimental evidence. This needs much more detail as GFP fluorescence as a proxy for germination potential is the central observation in this study.”

This is an important question that the new Figs. 4-7 address. We have performed a number of additional experiments that led to the conclusions stated on pgs. 1-3 of this rebuttal letter. We found that any two of the following three are highly and positively correlated with each other: GFP inducibility, amount of RNA polymerases (all three, RNAPs I-III), and germination ability. The RNAPs - we focused on RNAP II since it makes all coding RNAs and some non-coding RNAs - now give us a mechanistic window into what GFP inducibility represents. Overall, the results (stated on pgs. 1-3 of this letter), show that spores must maintain gene-expressing ability during dormancy in order to germinate, but that as they age, the spores gradually lose this ability because they gradually have less amounts of molecules that are required for gene expression, including RNAP II (we measure RNAP II production and degradation rates inside dormant spores and show that RNAP II level inevitably decreases because the production is too slow compared to degradation).

“(4) As previous studies indicate yeast spores continue metabolic activities while incubated in 30C water, the fact that dormant spores slowly cease to germinate seems to be a result of the experimental setup, not a general characteristic of dormancy as claimed in the introduction (Lines 69-73). This limitation was not clear from the manuscript. In addition, the statement on yeast spores retaining germination ability for “several months” in 4C water (Lines 649-651) needs more detail. Perhaps repeating the experiment in a much lower temperature may yield insight.”

We can indeed keep yeast spores - and vegetative yeasts - at different temperatures and can even freeze dry them. Indeed, different ways of keeping yeast spores likely lead to different lifespans since, for example, thermodynamics dictates that that biomolecules inside cells (and spores) must thermally degrade faster at higher temperatures than at lower temperatures - net loss of molecules that are required for gene expression is responsible for the ageing and death of spores as our additional experiments show. Indeed, in our experience, we found that we could keep our yeast spores alive in the fridge (~ 4 °C) such that about 20% of the spore population could still germinate after one year of being kept in the fridge. This indicates that spores can avoid death for much longer time at 4 °C than at the 30 °C that we used for all the reported experiments in our manuscript. We picked a particular set up for ageing dormant spores, 30 °C (conventional temperature in which one studies *S. cerevisiae*) in PBS or water. This is the most generic setting

that one can think of for *S. cerevisiae*. Our setup still demonstrates how dormant yeast spores approach their deaths even though we did not repeat all our experiments at multiple different temperatures. Repeating all our experiments at different temperatures is beyond the scope of our current work and is unnecessary for the main story.

"5) Line 430: "Dormant spores would gradually lose their ability to express genes without nutrients due to, for example, their intracellular components naturally (thermally) degrading over time" is a statement that is unsupported by evidence in the manuscript and too broad. Is this a general characteristic of microbial spores? Is there evidence that internal components are thermally degraded?"

This argument - that biomolecules will eventually degrade - follows from thermodynamics. But more importantly, we performed experiments during the revision that demonstrates this (e.g., measured the half-life of RNAP II as ~10 days and show evidence of other molecules that are required for gene expression degrading in dormant spores, some on the order of ~12 hours).

"(6) The study asserts that spores unable to germinate in saturating (2%) glucose are dead. This is a strong statement, and needs further clarification. Are there any known germinants other than glucose? What happens when you add another pulse of 2% glucose, much like the experiments in Fig 2? Are they unable to form colonies on rich media agar plates?"

The fact that glucose is required for germinating a yeast spore is well established in the literature. For example, one of the papers that we cite (Joseph-Strauss et al. *Genome Biol.* (2007)) states: *"We find that the germination process can be divided into two distinct stages. During the first stage, the induced spores **respond only to glucose**. The transcription program during this stage recapitulates the general transcription response of yeast cells to glucose. Only during the second phase are the cells able to sense and respond to other nutritional components in the environment."*

Other papers support this finding that glucose is required for germination (and giving the saturating level (2%) glucose is the what all yeast spore papers use to measure the % alive versus % dead). Amino acid supplements are required as well but without glucose, spores cannot germinate (and we have tested this and included this data in our manuscript - this is the "0% glucose" condition, in fact).

"Minor Points

1. A more thorough description of the yeast spore would be desirable. The current introduction makes broad associations with results from bacterial spores, which makes understanding the results in this paper more difficult."

We have now revised the introduction and discussion sections to restrict ourselves to yeast spores, while still giving some background info on microbial (including bacterial) spores because we believe that it is important for readers to place our work in a broader context of microbial (eukaryotic and prokaryotic) spores. But as we stated before, compared to the literature on dormant bacterial spores, there are much less papers that examined the dormant state of yeast spores itself (thus the relatively few citations on such papers compared to bacterial spores).

"2. Figure 1b needs a scale bar and time stamps."

We now state the duration of time between frames (10 minutes) in the caption. All pictures in the manuscript contain a single spore bag. Hence a scale bar for one picture indicates how big every spore bag in the picture are. Almost all the microscope images have a scale bar. For the minority of the pictures, we did not add the scale bar because these pictures they are either duplicate of the pictures next to them, but in a different fluorescence channel, or they already contain texts that indicate other features (when we tried, the pictures became unintelligible with all the texts on top of them).

"3. Figure 2: The results on primed dormancy do not have a strong association with the rest of the manuscript and seem to be a sidebar. Reorganizing it to put more emphasis on the motivation of the experiment (showing that ungerminated spores are not necessarily dead) would streamline the paper. "

We kept this part in the main story. While we do not return to primed dormancy after Fig. 2, we want to show that spores that do not germinate the first time can still germinate if they are given more glucose, which is connected to the points that we make in the later figures. We now emphasize even more that the point of two-pulse glucose experiment in Fig. 2 is - as the reviewer correctly points out - to show that un-germinated spores are not necessarily dead.

"4. Figure 2b Y-axis: is this a percentage of the total spores measured? The numbers do not add up with the plot in 2c upper row. "

Thank you for pointing this out. The y-axis of Fig. 2B should actually say "Number of germinated spore bags" (it reports how many spore bags germinated within the binned time windows on the x-axis). We have now corrected this.

"5. Fig 3c: the two plots should have matching Y-axis ranges."

We tried. But if we do this, then all the curves in the upper graph will get squished down on the x-axis and one cannot see the variability between the traces (one cannot see the data). To show the data and the variability, we kept the axes to be different (zoomed in for the upper panel).

"6. The paragraph starting from line 344 is perhaps too dramatic and can be omitted."

We edited this.

"7. Line 362: 'rather than a qualitative relationship such as 'spores that can express GFP can germinate whereas those that cannot express GFP do not germinate'.' is unnecessary. "

We modified this sentence.

"8. Line 568: comparison to plant seeds is a stretch because seeds pack nutrients that warrant initial growth. "

We removed the comparison to plant seeds.

"9. There are many instances of grammatical errors throughout the manuscript, which do not necessarily take away from the substance of the paper, but will benefit from a professional editing service."

One of the authors, whose primary language is English, went through the manuscript again to correct any grammatical errors but such errors may still remain. We will work with the copyeditor at the journal to correct these.

We thank Reviewer 1 for his/her comments. They have substantially improved our manuscript.

Response to Reviewer 2:

We thank Reviewer 2 for his/her positive overall assessment. Below, Reviewer 2's remarks are in *italics* and are followed by our responses.

"Summary

This study focuses on the dividing line between dormancy and death. Using yeast spores as a model for dormancy, the authors explore how dormant lifeforms can either be revived or be unrevivable. Combining systems and synthetic biology approaches, they identify how a spore's gene expression level is related to the level of glucose required to germinate later and how likely germination is instead of death."

We thank the reviewer for this positive overall assessment.

"General Remarks

This interesting study focuses on defining death vs. dormancy and what factors differentiate dormant spores that can and cannot be revived. This is an exciting and relatively unexplored question and the authors take a very creative approach to tackling this conundrum. The study's key finding that some global control mechanism of transcription and translation is linked to revival potential for spores is very exciting and will be of broad interest to the microbial research community. However, one concern is that the study design may potentially conflate the ability to sense the environment while dormant (sensing doxycycline and glucose) with the ability to express GFP when dormant and revival ability. While these factors are highly linked, because if you cannot sense the environment you cannot receive the signal to express GFP or be revived by glucose, some commentary by the authors on this point would strengthen the manuscript. For example, the priming phenomenon discussed might be that sensing some glucose induces spores to upregulate their ability to sense environmental changes in general. This could assist with their eventual revival and shorten the time required to sense and respond to these changes, leading to a shorter revival time."

We thank the reviewer for this positive overall assessment. We completely changed the discussion section, which now addresses the reviewer's comment about whether or not dormant yeast spores sense their external environment (e.g., sensing extracellular doxycycline and extracellular glucose) and, if so, how such sensing might play a role in the *GFP* expression. In short, yeast spores do not sense extracellular doxycycline. Doxycycline is a small molecule that readily diffuses into a spore through the spore wall, just as doxycycline readily diffuses into a vegetative yeast through

the cell wall. Yeasts do not have transporters or membrane-bound sensors for doxycycline. After diffusing into the cell, doxycycline binds to a transcription factor (rtTA in Fig. 3a) which then binds to the *TET* promoter to activate the *GFP* expression. Hence there is no active sensing or transport - no ATP-consuming process - involved in activating the *GFP* expression in our synthetic circuit.

Regarding whether dormant yeast spores sense extracellular glucose: *S. cerevisiae* has membrane-bound receptors, Snf3 and Rgf2, that sense extracellular glucose. The sensors bind extracellular glucose, without transporting them into yeast, to determine the concentration of extracellular glucose. Specifically, the sensors trigger an intracellular signal cascade that regulates the expression of seven passive transporters for glucose, Hxt1-7. Since we induce the *GFP* expression in the dormant yeast spores in water or PBS, without any external glucose and other nutrients, there is no glucose for the spores to sense that would affect the *GFP* expression and our interpretations of the results in Fig. 3 regarding the synthetic circuit. In the priming experiments, sensing of extracellular glucose, along with the effects of intracellular glucose brought in by the Hxts1-7, likely affects the expression levels seen in the RNA-Seq data set (Fig. 2). The reviewer indirectly touches on an interesting topic in his/her comment: is a dormant yeast spore actively (using ATPs) and constantly sensing its external environment? We don't have evidence for this. If they do, then what would be the best strategy for sensing the environment in a way that maintains the stored ATPs for as long as possible? We believe that this question invites future investigations.

"Minor points

- Abstract (lines 33-34) and other locations: "How organisms with ceased lives remain alive and what sets their lifespans are fundamental questions." The idea of ceased life being the same as dormancy appears here and in other locations throughout the manuscript. The idea of ceased life implying dormancy and not death could be confusing to readers who think of ceased life as death. It may be helpful to explicitly define dormancy, ceased life, and death in the manuscript as these terms are used frequently and have specific meanings that are explored and challenged throughout the text."

We have now revised the text to clarify these points when we introduce them in depth. For example, we define a dead spore in the results section, just before we measure the percentages of spores that are alive (just after Fig. 3). We have also revised the introduction to explain how challenging it is to clearly define "dormancy" (and this is a motivation for our study indeed) and remind the readers of this challenge in the discussion section (i.e., how silent the state of dormancy really is the question that we largely answered in yeast but still many aspects of this question remain).

"- Lines 197-199: "These results establish that spores that do not germinate after encountering

ample glucose are not necessarily dead." The timing of the ample glucose exposure should be clarified in this sentence as this section is about priming germination with an initial encounter with glucose and then a second, subsequent encounter. "

We have now indicated the timing of the glucose exposures in these and surrounding sentences.

" - The section "Synthetic circuit shows that activating GFP expression in dormant yeast-spores does not alter percentages of spore bags that germinate" discusses an important control for the synthetic biology approach to understanding what controls the ability of dormant spores to germinate, but may be best moved to the supplement. "

We thank the reviewer for this suggestion. But after various rewrites, we decided to keep this part in the end because it describes a figure that disproves our original hypothesis, which is that making of GFP - and spending energy to do so - does not diminish or alter the spore's germination ability. We feel that this is important for the flow of the story because the hypothesis, despite it being wrong, was the motivator for introducing the GFP-inducing synthetic circuit. So we think that this data should be in Fig. 3 as an evaluation of the hypothesis.

" - It could be exciting in follow-up work to explore the implications of priming as a phenomenon specifically to see if arbitrary gene expression ability changes after priming. This is implied by the upregulation in gene expression across most genes in Figure 2g, but isn't explicitly revisited after the synthetic GFP expression circuit is introduced. "

In the new discussion section, we now include this point as one of the future studies that the study encourages.

"- General comment: The way Glucose % is displayed in figures should be consistent throughout the manuscript. For example, in Figures 1 and 2, Glucose % is displayed in scientific notation, but in Figure 3, decimal notation is used."

We tried to be consistent except for the few cases where putting the more bulky scientific notation (due to the superscript) made the figure too crowded. In the end, we certainly use the scientific

notation when the axis spans many multiples of 10s (e.g, 10, 100, 1000, etc.) but in cases where the values change by a few folds (e.g. 0.001, 0.002, 0,003) and the scientific notation is too bulky for the figure, we decided not to use the scientific notation.

We thank Reviewer 2 for his/her positive overall assessment. His/her comments have improved our manuscript.

Response to Reviewer 3:

We thank Reviewer 3 for his/her constructive comments. We also thank him/her for detailed suggestions on how to look for more molecular/mechanistic explanations of the main phenomenon in our original manuscript: GFP inducibility predicting the germination ability. His/her suggestions greatly helped us - we performed experiments that we would not have thought of ourselves - and we have performed almost all the experiments suggested by the reviewer. Afterwards, we developed follow-up experiments ourselves to further explore the outcomes of the experiments suggested by the reviewer. We have now obtained more molecular/mechanistic explanations that have substantially improved our manuscript. Below, Reviewer 3's remarks are in *italics* and are followed by our responses.

"Spores are typically thought to be in a state of metabolic inactivity, where a glass-like cytoplasm preserves cellular integrity. This results from both direct evidence in some species and the fact that spores can remain viable for many years; thus, it seemed unclear how any spores that had metabolic activity could remain viable. Contrary to this, a small amount of work has suggested that, at least in some species, spores have metabolically activity. The authors here convincingly show a surprising new aspect of the spore physiology: yeast spores are able to express genes to a significant level, as shown by the expression of GFP from a synthetic construct. Even more surprising, the GFP expression level correlates with the probability of a spore to germinate. With this system, the authors characterize the relationship between germination requirements and GFP expression. They find that yeast spores require a certain concentration of glucose to germinate. When expressing GFP from an inducible promoter, higher GFP expression levels lower the concentration of glucose required to sporulate. In addition, over time they find that spores express less GFP, and cells require higher glucose concentrations to germinate. From this finding, the authors draw the conclusion that yeast spores lose the ability to germinate, because they eventually require a concentration of glucose higher than what they can be exposed to (>2%)."

We thank the reviewer for this positive assessment of our findings.

"The interesting phenomenon observed by the authors, that GFP expression correlates with the probability to germinate, by itself, I believe, is insufficient for publication in Molecular Systems Biology. Several of the figures in the manuscript, such as the double pulses of Fig. 2 or the model of Fig. 4 do not help in strengthening the key point of the paper. The minimal amount of glucose needed to germinate (Fig. 1&2) and the ability to express GFP (Fig. 3&4) are not causally linked in this manuscript, other than by a model that is mathematically describing the correlation. Taken together, I cannot currently advice publication of the manuscript."

During the past nine months, with some limitations imposed on the working hours due to COVID-19, the first author (Théo Maire) diligently performed additional experiments (Figs. 4-7) that led to a molecular/mechanistic explanation of why the inducible *GFP* expression level ("GFP inducibility") highly and positively correlates with the spore's probability of germinating ("germination ability") for each glucose concentration. To describe the additional work, we eliminated the old Fig. 4 and replaced it with the current Figs. 4-7. We explain the new results in more detail in our responses below.

Before doing so, we explain here why we decided to keep Fig. 2. The main point of Fig. 2 is that a spore is not dead just because it does not germinate after receiving less than the saturating concentration (2%) of glucose. The fact that not every spore germinates - subject of Fig. 1 - has never been reported in the literature, to our knowledge, because no published works on dormant yeast spores used glucose concentrations smaller than 2%. The RNA-Seq data in Fig. 2 also provides an insight that we missed before, now in light of the additional work in Figs. 4-6 (explained in the text that explains Fig. 6). So, while Fig. 2 is unnecessary for understanding Figs. 3-6, it is important for our story to show that increasing the glucose concentration induces germinations of the spores that did not germinate the first time (i.e., showing that they were still dormant rather than dead). Moreover, in light of our claim that dormant spores can and do express genes (now shown concretely by, for example, showing that RNAP II is actively being produced during dormancy), the RNA-Seq performed on the primed, un-germinated spores in Fig. 2 is important because it shows that dormant spores indeed can express genes other than *GFP*. For these reasons, the revised manuscript still contains the same Fig. 2 despite the fact that one can go deeper and explore primed dormancy further, which we did not do in order to stay on message.

"I do believe, however, that by probing what the GFP expression ability means, why it is decreasing and why this is important for maintaining viability, the authors could eventually bring this manuscript on a level appropriate for Molecular Systems Biology. The basic phenomenon reported in this work is interesting and important for the community, and I encourage the authors to probe deeper for the molecular mechanism. I provide some feedback below. I do not expect the authors to perform all of these experiments, and I do not know, if all experiments are performed, whether this manuscript will be sufficient for publication in Molecular Systems Biology as it is hard to predict the results. If the authors decide to resubmit this paper, I would be happy to re-review this manuscript."

We thank the reviewer for the encouragement and suggesting new experiments. We have now performed almost all the suggested experiments and also follow-ups that we designed. The only major suggestion that we did not follow is further exploring the phenomenon of primed dormancy. We believe that this will detract us from the main goal - determining how yeast spores age and die during dormancy.

The suggested experiments and their follow-ups allowed us to mechanistically explain what underlies the GFP inducibility and its connection to germination ability, which decays over time. Namely, we found that the yeast spores gradually lose molecules that are required for gene-expression, including RNAP II, as they age and that eventually, the spores die because they have irreversibly lost the gene-expressing ability. Here, irreversible means that even after receiving a 2%-glucose, the spores are not able to express genes, including those required for replenishing the RNAP II that has depleted to an undetectable level. Gene expression is necessary for germination because a spore needs to build a new cell that will bud off of it (i.e., germinate). Having lost sufficient amounts of the molecules that are required for gene expression during dormancy, the spores cannot recover these after receiving a 2%-glucose and thus fail to germinate (they are dead). We put either firm numbers or timescales for the rates at which dormant spores lose the different molecules that are required for gene-expression (e.g., RNAP II is gradually lost with a half-life of ~8 days while some other key molecules - which we argue are likely transcripts that are important for gene-expressing ability - degrade with a timescale of hours).

In our responses below, we elaborate on the additional experiments and conclusions.

"Major points

1) The authors' argument is based on two correlations. First, GFP expression levels correlated with the minimum glucose concentration required for sporulation. Second, GFP expression levels anticorrelated with time. Combining these two correlations will result in a new correlation: the GFP expression ability at one point in time will correlate with its life time. While plausible, the authors would need to show 1) that spores with a higher GFP expression ability will actually live longer (I suspect that the authors are right, but they need to provide experimental evidence for this claim); and 2) the authors would need to show experimental evidence that glucose is limiting, as the authors claim that spores lose the ability to germinate because they cannot be exposed to high enough glucose concentrations."

To address the reviewer's two points regarding ageing, (1) & (2), we performed additional experiments described in the new figures, Figs. 5-7. We will explain these in detail in our responses to the reviewer's subsequent comments below. But in short, since our manuscript no longer uses the mathematical analysis of the two correlations for claiming that spores with more GFP live longer, we no longer need to sort the spores with more GFP-inducibility and then show that the mathematical analysis is true. We took a different route to address the central issue concerning points (1) and (2): the intracellular workings that take place over days to weeks as a dormant spore ages towards death (and one that corresponds to the GFP inducibility gradually being lost). We piece together several experiments to reveal this picture. In one of them, we have engineered new yeast strains in which a subunit of RNA polymerase II (RNAP II) is fluorescent (tethered to mCherry), sporulated this strain, and then performed an additional "ageing experiment" in which we starved the engineered spores in water without any nutrients (Fig. 5). This experiment allowed us to track RNAP II level in dormant spores as they approach their death

over ~2 months. Through this and related experiments that we performed during the revision, we now give a more mechanistic view of ageing during dormancy and spore death.

"2) Why does higher GFP expression lead to higher germination rates? This is the central point where the manuscript must be improved to be publishable in a journal like Molecular Systems Biology. Understanding what GFP expression ability is a proxy for can give the authors valuable insights into why spores are dying. Some progress needs to be made. One option is that GFP expression ability correlates with the internal energy state of the spore. One way to test this, could be to add ionophores like DNP or CCCP to inhibit ATP production and to see if their presence during dormancy will shorten the life span of the spores. Another approach could be to block transcription/translation and test if protein synthesis during dormancy is important for survival. Adding nutrients, glucose or other carbon substrates, could replenish the energy pool (if limited), and increase GFP expression. Will it also increase life span? In case spores are energy-limited, expressing luciferase prior to sporulation could allow the authors to get a direct measurement of the ATP content in the spores."

We performed these experiments (Fig. 4A-B and associated supplementary figures). Namely, we used four drugs - antimycin A, CCCP, cycloheximide, and thiolutin. Antimycin A and CCCP act equivalently. They both inhibit the ATP synthase by blocking proton transport across membranes (thereby inhibiting oxidative phosphorylation). Since both drugs gave the same results, we reported only the results for antimycin A. Cycloheximide globally inhibits translation. Thiolutin globally inhibits transcription. For each drug, we kept dormant spores in water with the drug for 24 hours. We then washed away the drug and replaced the water with minimal medium with a 2%-glucose (saturating level of glucose) (Fig. 4A). This experiment yielded a striking, yes/no-type answer (Fig. 4B). Blocking ATP synthesis for 24 hours hardly killed any of the spores: nearly 100% of the spores still germinated after receiving a 2%-glucose just like a control population that we kept in water for the same amount of time without any drugs. Blocking global translation for 24 hours also hardly killed any of the spores. But blocking global transcription for 24 hours resulted in nearly every spore dying - almost no spore germinated after receiving a 2% glucose. These experiments exclude the idea that ATP / internal energy state being the primary determinant of germination ability or that the leading cause of death is spores not replenishing the stored ATPs after depleting them, at least on a timescale of a day. Hence, we have not performed some of the follow-up experiments that the reviewer mentions above such as giving different carbon substrates for replenishing the intracellular energy pool and the luciferase-based experiments.

To understand why inhibiting transcription for 24 hours kills almost all spores, we measured the abundances of RNA polymerases (RNAPs) I-III in individual spores. Specifically, we genetically engineered spores so that a fluorescent protein, mCherry, is tethered to Rpb3 - a major subunit of RNAP II. With time-lapse microscopy, we then quantified the abundance of RNAP II in individual spores by measuring the mCherry fluorescence in each spore as a proxy (Fig. 4F). Likewise, we quantified the abundance of RNAP I & III combined in individual spores by genetically

engineering a strain that expresses mCherry tethered to Rpc40, a subunit shared by RNAP I and RNAP III (we could not find a workable subunit that was unique to RNAP I but this isn't crucial for our conclusions) (Fig. 4G). With these measurements, we found that if a dormant spore had more RNAP II, then it had a higher GFP inducibility in water/PBS (i.e., higher steady-state level of GFP reached, after 24-hrs of doxycycline) (Figs. 4F-G). This result manifests as a statistically significant, highly positive correlation between the GFP inducibility and RNAP II level (Fig. 4F - $R=0.64$, p-value in caption). Likewise, we found a statistically significant, highly positive correlation between the GFP inducibility and the combined level of RNAP I & III - the two RNAPs that transcribe only non-coding genes (Fig. 4G - $R=0.63$, p-value in the caption). Since a dormant spore with a higher GFP inducibility is more likely to germinate in less-than-saturating concentrations of glucose, we found that spores with more RNAP II were more likely to germinate (we focused on RNAP II to constrain the follow-up experiments to a reasonable amount)(Figs. 4I-J). Unlike the RNAPs, we found a statistically significant but weakly positive correlation between the GFP inducibility and the abundance of ribosomes which we proxied by the amount of 18S rRNA that we measured with single-molecule FISH in individual spores (Fig. 4H). This is consistent with the finding that a 24-hour inhibition of global translation had virtually no effect on the spores' survivability (i.e., germination ability at 2%-glucose) (Fig. 4B).

We thus address the reviewer's key question - what is the GFP inducibility a proxy for? - in the following way. The GFP inducibility is a proxy for the abundances of RNA polymerases (RNAP I, II, and III). Then the question becomes what the RNAP II level represents in a spore and why it is highly and positively correlated with the spore's germination ability for a given glucose concentration (Fig. 4I). The new experiments in Figs. 5-7 address this question. We first show that the RNAP II - which we focused on rather than RNAPs I & III - become less abundant in dormant spores as they age in water without any nutrients (Fig. 5C). Concurrently, the aged spores have less GFP inducibility as expected (Fig. 5B). The aged spores, with less RNAP II than the younger spores, are more likely to be dead (Fig. 5D). Furthermore, moments before dying - after many days (e.g., 39 days) of being in water without nutrients at 30 °C - the dormant spores have barely detectable RNAP II levels just like dead spores (Fig. 5E) but, unlike the dead spores, they are able to make more RNAP II after receiving a 2%-glucose (Fig. 5F). That is, they still can express genes after receiving the 2%-glucose. Gene-expression is necessary for germination since the spore would need to make a new cell that will bud off (i.e., germinate). Hence if the spore cannot express genes even after receiving a 2%-glucose, it cannot germinate (i.e., it is dead). Indeed, we have not found any dead spore that is able to make more RNAP II or express GFP after receiving a 2%-glucose. These results suggest, and supported by further experiments, that spores lose the ability to express genes moments before dying.

But germination ability does not depend on the RNAP II level alone. Rather, to address the question of what the RNAP II level really represents, it is one of several measures of the gene-expressing ability that a spore has during dormancy. Indeed, we establish that gene expression does occur during dormancy, not only when it is forced to as in the *GFP* expression (e.g., dormant spores actively synthesize RNAs (Figs. 4D-E) and RNAP II (Fig. 7)). We also show that spores with abundant RNAP IIs can still have no gene-expressing ability even after receiving a 2%-glucose and are thus dead if their global transcription was inhibited for 12 hours before receiving the 2%-glucose (Figs. 6C-E). This result suggests that some molecules, which are required for

gene expression, were degraded during the 12 hours of transcription inhibition. We argue that these molecules are likely transcripts though we also state that further investigations - beyond the scope of our manuscript - are required to identify the transcripts and solidify our claim here. Our claim is partly based on the fact that RNAP II degrades on the time scale of ~8 days (half-life measured in Fig. 7B). Hence, we argue that protein-based machines like RNAP II would degrade on the same timescale. We also find many ribosomal RNAs after transcription inhibition, thus we suspect that these transcripts are likely not rRNAs (besides, the 18S rRNA - a subunit of ribosome - poorly correlating with GFP inducibility (and thus germination ability) argues against ribosomes being the limiting factors for germination). In the end, we give our model for dormancy-to-death transition by piecing together all the data (Fig. 7F). The main idea of the model is that a dormant spore maintains active gene expression as exhibited by, for example, them producing new RNAP IIs. But the spores are producing the molecules that are required for gene expression more slowly than the time taken by the molecules to degrade. Thus, there is a net loss of these molecules, with different classes of molecules (e.g., transcripts and RNAPs) degrading at different rates. This is what it means for the yeast spores to age during dormancy. After the spores are sufficiently old, they have lost enough copies of at least one of the molecules that are required for gene expression. After this time, the spore can no longer germinate even after getting a 2%-glucose because it is not able to express genes (gene expression is required to make more of the molecules that it had lost and also to build a new cell). As a result, the spore is dead.

The main mystery is what is being lost during the transcription inhibition - we argue that these are key transcripts, perhaps non-coding RNAs. But we cannot address this question, we believe, without another manuscript's worth of work. Our revised manuscript hence addresses the reviewer's question, by invoking explanations that are more mechanistic than the ones that we gave in the original manuscript, but also raises new questions.

" 3) Directly measuring life span as a function of GFP expression levels. The authors have good evidence that the GFP expression ability at given point in time correlates with their ability to germinate. The authors are missing an experiment that shows that higher GFP expression predicts also correlates with germination efficiency at a later time. The authors could induce GFP expression for a limited time, e.g. for 1 day early in sporulation, and then measure the germination probability several days later. "

We did not perform this experiment for the reason given in our previous response - we took a different route (new Fig. 4 and Figs. 5-7) that negates the need to experimentally confirm the mathematical analysis of the lifespan based on GFP inducibility that we previously presented the old Fig. 4. The original conclusion is still true but now we provide the mechanistic explanation given in our previous response.

"4) Model of the increase of minimal glucose concentration. According to the calculation of Fig. 4f, spores with low GFP expression ability increase their minimal glucose concentration at a rate of less than 1.5 10⁻³%/d. At this maximal rate, after 40 days, I expect the minimal glucose concentration to be 40d x 1.5 10⁻³%/d = 0.06%, well short of the 2% maximal glucose concentration of the experiments. This is obviously not what the authors' model produced. The authors' model seems instead to be built around the decrease of GFP expression ability from a finite number down to zero. Once the cells hit zero, see Fig. S24d, spores are pronounced dead. The model is missing an obvious test: does the estimated life span of Fig. 4g correctly predict the survival curve of Fig. 4a? Apart from this, the model makes sense, but it took me an awful long time to understand it. If I interpret it correctly, with the model the authors can discuss the kinetics of death. An aspect that is hidden in the experimental data of Fig. 4c, which only considers viable spores. It was unclear to me, why the authors dwell on the minimal amount of glucose to wake-up, Fig. 4f. The loss of GFP expression ability seems to be clearly correlated with survival and a much more physiological meaningful parameter. The minimal amount of glucose to wake-up diverges at the lowest GFP expression ability in Fig. 4e, making it hard to measure and hard to interpret. It is furthermore not clear whether the model is truly describing the minimal amount of glucose to wake-up. Rephrasing the model in terms of GFP expression ability could help it describing a more meaningful aspect of the spores' life. "

The main conclusions from the original manuscript, including on ageing, are still true. But we now have a different storyline. We now focus on the molecular mechanisms underlying the GFP inducibility to study the dormancy-to-death transition. Thus, we removed the mathematical analysis of the GFP level decreasing over time and the associated figures (this was the old Figure 4 which has now changed and is followed by two new figures, Figs 5-6). We have performed a new "ageing experiment" in which we starve the newly engineered spores - those with RNAP II tagged with mCherry - in water without nutrients over ~2 months. Figure 5 now reports this new experiment in addition to the ageing experiment that was in the original Figure 4A (just watching GFP inducibility decrease over time). With the new storyline, the mathematical analysis - while still valid - is no longer relevant. Thus, we have not performed the above experiment proposed by the reviewer.

"Minor points

1) Priming with low glucose concentration. Fig. 2 seems misplaced in the story of the paper. Why prime the spores with glucose, but draw no conclusion from the result? I would suspect that the GFP expression ability changes after a low glucose pulse. If the GFP expression ability changes, the authors have a very beautiful system at hand to test if the lifespan of the spores, changes,

also. "

We kept this part in the main story. While we do not return to primed dormancy after Fig. 2, we want to show that spores that do not germinate the first time can still germinate if they are given more glucose, which is connected to the points that we make in the later figures. We now emphasize even more that the point of two-pulse glucose experiment in Fig. 2 is - as the reviewer correctly points out - to show that un-germinated spores are not necessarily dead.

"2) Are all yeast sporulated. I assume that the authors have a way to ensure that all yeast are indeed sporulated, or at least that only sporulated yeast are included in the data analysis. I would like to see evidence of this."

The conventional sporulation protocol causes most of the vegetative yeasts to sporulate (we visually see this with our microscope). But it does not ensure that every yeast within a starved population sporulates. Some of the vegetative cells die but maintain an intact shape while some others cease to grow but without dying or sporulating. But this is okay because we used microscopy with single-cell resolution to only track and study spores, not vegetative cells that were still alive (and thus will divide after we give glucose) or cells that died without sporulating (and thus will not divide after we give glucose). Almost all of our experiments used microscopy with single-cell resolution so we could clearly distinguish spores from vegetative cells (e.g., spores are contained in spore bags whereas vegetative cells are alone). As mentioned, a large majority of the population has sporulated.

For the few bulk (not single-cell) experiments, namely for the RNA-Seq (Fig. 2), we gave zymolyase to the entire population after sporulation. This standard technique for isolating spores - useful for bulk but not single-cell experiments - lyses the vegetative cells and leaves spores almost intact (Supplementary Fig. 7). But it also damages the spores in that the zymolyase degrades the spore wall. This was okay for the RNA-Seq since we would next lyse these spores anyway to extract their RNAs. But for our functional studies with a microscope - such as giving glucose to check spores' germination ability - we cannot cleanly exclude the possibility that the lysis of the spore wall causes the spores to have either higher or lower chances of germinating. We thus believe that our approach, which visually selects and tracks individual spores with a microscope, is the most non-invasive method that is suitable for our functional studies (i.e., checking how well each spore germinates).

"3) Glucose concentration range. In experiments such as Fig. 4c, the authors show a very narrow concentration range, while Fig. 4e shows a different range. Changing glucose concentration ranges makes it hard to interpret data (e.g. from 4c to 4e). Why the authors focus on the range of 0.001% to 0.01% to make the point that the minimal glucose concentration to germinate is increasing above 2% is confusing. What about the intermediate regime?"

Since our storyline has changed, we have completely revised Fig. 4. These plots in Fig. 4 are thus no longer present and unnecessary. This resolves the issue.

"4) 99% germination metric. From germination landscapes like Fig. 4c, the authors extract the minimum concentration of glucose to wake up 99% of spores. It is unclear how the authors got this metric. A fit of a mathematical model? Or the lowest experimental concentration where more than 99% sporulated? Either way, this seems like a dangerous metric, that could depend on the detailed choice of the model, or the experimental noise. The authors should show that their results are robust to other choices of metrics."

Since our storyline has changed and now includes the molecular mechanisms that underlie GFP inducibility, we no longer need the mathematical argument. We have thus removed the mathematical analysis. This resolves the issue.

"5) Synthetic promoter. Spore bags induced with doxycycline plateau at a certain expression level. It is a little surprising given that there is no cell growth that level don't continue to increase over time. Doxycycline is unstable and potential is functional depleted in the experiments by the end of one day. It would be good for the authors to swap media with fresh doxycycline once per day to ensure their results are not influenced by doxycycline breakdown."

The GFP protein level plateaus after the 24 hours of doxycycline because the production of GFP shuts off. This is despite the saturating concentration of doxycycline (100 ug/ml) remaining virtually undiminished during the 24 hours (Supplementary Fig. 14). At the incubation temperature of 30 °C, doxycycline has a half-life that is longer than at least two days as we verified in Supplementary Fig. 14. The reviewer is correct that doxycycline can be unstable. But this is only in some conditions, most notably when there is light. Our experiments are done in a dark room with relatively infrequent and short (millisecond) flashes of light to measure GFP levels over time. The reviewer is also correct that the GFP level plateauing in the spores, which do not divide or grow in volume, is surprising. As we reported in the original manuscript (Supplementary Fig. 13), this means that the production of GFP has halted before the end of the 24 hours of doxycycline. This

surprising phenomenon reflects the interesting and poorly understood nature of transcription and translation in dormant yeast spores, which invites future investigations into the central dogma of molecular biology in dormant yeast spores.

We thank Reviewer 3 for his/her constructive comments and highly insightful suggestions. The more mechanistic insights that we revealed during the revision have strengthened our manuscript. The fact that we were able to achieve was largely due to this reviewer. We believe that our manuscript now has a broadly interesting, systems-level phenomenon (established prior to the revision) that is now supported by mechanistic/molecular insights (discovered during the revision). We hope that our revision has sufficiently addressed most of the reviewer's comments.

Thank you for sending us your revised manuscript. I apologize once again for the delay in getting back to you, which was due to the late arrival of the referee reports. We have now heard back from the three reviewers who were asked to evaluate your revised study. As you will see below, they think that you have done a great job with the revisions and they are supportive of publication. It is really nice to see such enthusiastic referee reports. I am glad to inform you that your manuscript is now suitable for publication, pending some minor editorial issues listed below.

We would ask you to address the following.

REFeree REPORTS

Reviewer #1:

The authors have done a solid job in addressing my comments (and as far as I can tell, they have also addressed comments from the other reviewers).

Given the restrictions faced by all experimental labs during this Covid pandemic, I suggest that no further revisions are needed and I support publication of this exciting work that sheds a new light onto the mysterious properties of yeast spores and their germination propensity. Congratulations to the authors on this interesting study.

Reviewer #2:

This revision is a real tour de force that digs deeply into the mechanisms underlying the transition between dormancy and death described in the original manuscript. Comments from all reviewers are fully addressed, new findings resulting from these concepts have been well-integrated into a clearly written revised manuscript, and I highly recommend publication.

Reviewer #3:

The revision is quite a tour de force. The new experiments really add a layer of depth and mechanistic understanding which has brought this paper to a whole new level. I commend the authors on the revision.

There are of course a number of interesting experiments that I could suggest to push this study even further (and I'm sure the authors have thought of many of them) but this is a paper not a book, and as a paper I think the story will now be intriguing to a wide readership.

The Authors have made the requested editorial changes

Dear Hyun,

Thank you again for sending us your revised manuscript. We are now satisfied with the modifications made and I am pleased to inform you that your paper has been accepted for publication.

Corresponding Author Name: Hyun Youk

Manuscript Number: MSB-19-9245